# Identification of compounds that rescue otic and myelination defects in the zebrafish *adgrg6* (*gpr126*) mutant

Elvira Diamantopoulou[1†], Sarah Baxendale[1†], Antonio de la Vega de León[2], Anzar Asad[1], Celia J Holdsworth[1], Leila Abbas[1], Valerie J Gillet[2], Giselle R Wiggin[3], Tanya T Whitfield[1]*

[1]Bateson Centre and Department of Biomedical Science, University of Sheffield, Sheffield, United Kingdom; [2]Information School, University of Sheffield, Sheffield, United Kingdom; [3]Sosei Heptares, Cambridge, United Kingdom

**Abstract** Adgrg6 (Gpr126) is an adhesion class G protein-coupled receptor with a conserved role in myelination of the peripheral nervous system. In the zebrafish, mutation of *adgrg6* also results in defects in the inner ear: otic tissue fails to down-regulate *versican* gene expression and morphogenesis is disrupted. We have designed a whole-animal screen that tests for rescue of both up- and down-regulated gene expression in mutant embryos, together with analysis of weak and strong alleles. From a screen of 3120 structurally diverse compounds, we have identified 68 that reduce *versican b* expression in the *adgrg6* mutant ear, 41 of which also restore *myelin basic protein* gene expression in Schwann cells of mutant embryos. Nineteen compounds unable to rescue a strong *adgrg6* allele provide candidates for molecules that may interact directly with the Adgrg6 receptor. Our pipeline provides a powerful approach for identifying compounds that modulate GPCR activity, with potential impact for future drug design.

DOI: https://doi.org/10.7554/eLife.44889.001

**\*For correspondence:**
t.whitfield@sheffield.ac.uk

[†]These authors contributed equally to this work

## Introduction

Adgrg6 (Gpr126) is an adhesion (B2) class G protein-coupled receptor (aGPCR) with conserved roles in myelination of the vertebrate peripheral nervous system (PNS) (reviewed in *Langenhan et al., 2016*; *Patra et al., 2014*). In homozygous loss-of-function *adgrg6* zebrafish and mouse mutants, peripheral myelination is severely impaired: Schwann cells associate with axons, but are unable to generate the myelin sheath, and show reduced expression of the *myelin basic protein* (*mbp*) gene (*Glenn and Talbot, 2013*; *Mogha et al., 2013*; *Monk et al., 2009*; *Monk et al., 2011*). Targeted disruption of *Adgrg6* in the mouse results in additional abnormal phenotypes, including limb and cardiac abnormalities, axon degeneration and embryonic lethality (*Monk et al., 2011*; *Patra et al., 2013*; *Waller-Evans et al., 2010*). In humans, mutations in *ADGRG6* are causative for congenital contracture syndrome 9, a severe type of arthrogryposis multiplex congenita (*Ravenscroft et al., 2015*). Peripheral nerves from affected individuals have reduced expression of myelin basic protein, suggesting that the function of ADGRG6 in myelination is evolutionarily conserved from teleosts to humans (*Ravenscroft et al., 2015*). Human *ADGRG6* variants have also been proposed to underlie some paediatric musculoskeletal disorders, including adolescent idiopathic scoliosis (*Karner et al., 2015*) (and references within).

In zebrafish, homozygous loss-of-function *adgrg6* mutants exhibit an inner ear defect in addition to deficiencies in myelination (*Geng et al., 2013*; *Monk et al., 2009*). In the otic vesicle, the epithelial projections that prefigure formation of the semicircular canal ducts overgrow and fail to fuse, resulting in morphological defects and ear swelling. Analysis of the zebrafish *adgrg6* mutant ear

shows a dramatic alteration in the expression of genes coding for several extracellular matrix (ECM) components and ECM-modifying enzymes (*Geng et al., 2013*) (*Figure 1A*). Notably, transcripts coding for core proteins of the chondroitin sulphate proteoglycan Versican, normally transiently expressed in the outgrowing projections and then down-regulated once projection fusion has occurred, remain highly expressed in the overgrown and unfused projections of *adgrg6* mutants (*Geng et al., 2013*). Although *Adgrg6* (*Gpr126*) mRNA is known to be expressed in the mouse ear (*Patra et al., 2013*), a role in otic development in the mammal has yet to be determined.

Like all aGPCR members, the zebrafish Adgrg6 receptor consists of a long extracellular domain (ECD), a seven-pass transmembrane domain (7TM), and a short intracellular domain (reviewed in *Langenhan et al., 2016*) (*Figure 1B*). The ECD includes a GPCR autoproteolysis-inducing (GAIN) domain, which incorporates the GPCR proteolytic site (GPS) and the conserved Stachel sequence (*Liebscher et al., 2014*; *Patra et al., 2014*). Proteolysis at the GPS results in two fragments, an N-terminal fragment (NTF) and a C-terminal fragment (CTF), which can remain associated with one another, or may dissociate, the NTF binding to cell surface or extracellular matrix ligands (*Patra et al., 2014*; *Petersen et al., 2015*). Dissociation of the NTF triggers binding of the Stachel sequence to the 7TM domain, thereby activating the CTF (*Liebscher et al., 2014*). This feature provides a variety of CTF-dependent or -independent signalling capabilities that orchestrate cell adhesion and other cell-cell or cell-matrix interactions. For example, during Schwann cell development and terminal differentiation, the Adgrg6 NTF promotes radial sorting of axons, whereas the CTF is thought to signal through a stimulatory Gα subunit (Gα$_s$), leading to elevated cAMP levels and activated protein kinase A (PKA) to induce transcription of downstream target genes, such as *egr2* and *oct6* (*Petersen et al., 2015*). Compounds that act to raise intracellular cAMP levels, such as the phosphodiesterase inhibitor 3-isobutyl-1-methylxanthine (IBMX) and the adenylyl cyclase activator forskolin, can rescue phenotypic defects in both the inner ear and PNS in *adgrg6* mutants (*Geng et al., 2013*; *Monk et al., 2009*).

Despite the enormous importance of GPCRs as drug targets (*Hauser et al., 2017*; *Sriram and Insel, 2018*; *Wootten et al., 2018*), adhesion class GPCRs remain poorly characterised, representing a valuable untapped resource as targets of future therapeutics (*Hamann et al., 2015*; *Monk et al., 2015*). The identification of specific modulators of aGPCR activity is an essential step for understanding the mechanism of aGPCR function and to inform the design of new drugs. Recent successful approaches include the use of Stachel sequence peptides as aGPCR agonists (*Demberg et al., 2017*), or synthetic monobodies directed against domains within the NTF (*Salzman et al., 2017*). A promising alternative approach lies in the potential of unbiased whole-animal screening of small molecules. In recent years, zebrafish have emerged as an important tool for in vivo phenotypic screening for new therapeutics (*Brady et al., 2016*) and for understanding biological mechanisms (*Baxendale et al., 2017*; *Richter et al., 2017*). Zebrafish have many advantages for drug discovery: they are a vertebrate species whose embryos can fit into individual wells of a multiwell plate, facilitating high-throughput analysis; they generate large numbers of offspring; they can absorb compounds directly added to the water, and whole-organism screening enables toxicity, absorption, metabolism and excretion of compounds to be assayed early in the screening pipeline.

To date, over one hundred drug screens using different zebrafish disease models have been conducted, some identifying lead compounds that have subsequently been tested in mammalian model systems or entered clinical trials (*Chowdhury et al., 2013*; *Griffin et al., 2017*; *North et al., 2007*; *Owens et al., 2008*) (reviewed in *Baxendale et al., 2017*). Two screens have been performed to identify compounds that promote myelination in the central nervous system (*Buckley et al., 2010*; *Early et al., 2018*). These studies used live imaging of *Tg(olig2:GFP)* or *Tg(mbp:eGFP)* fluorescent transgenic lines to screen for small molecules that increase progenitor or myelinating oligodendrocyte cell number. While elegant in design, and successful in identifying hit compounds, these screens required the use of sophisticated and costly high-resolution imaging platforms and relied on detailed quantitative assays for cell number, techniques that are not available to all labs and are potentially limiting in scalability and throughput.

In this study, we have developed an in vivo drug screening assay based on semi-automated in situ hybridisation (ISH) to identify modulators of the Adgrg6 pathway. We have used the otic expression of *versican b (vcanb)* as an easily-scored qualitative readout to identify compounds that can reduce *vcanb* overexpression back to normal levels in a hypomorphic mutant allele for *adgrg6, tb233c*. We used expression of *mbp* in the posterior lateral line ganglion of *adgrg6*$^{tb233c}$ mutants as a secondary

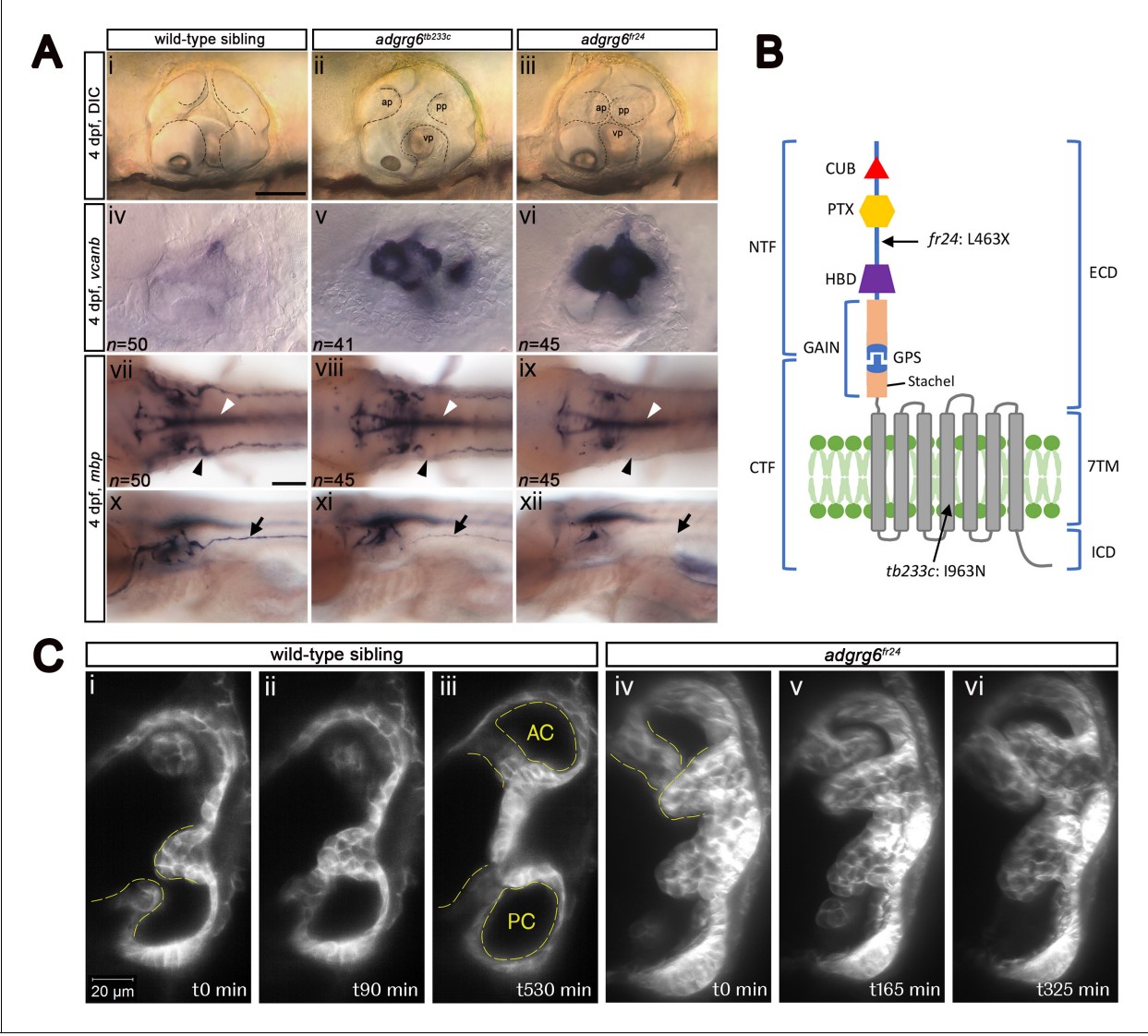

**Figure 1.** Comparison of *adgrg6* mutant allele phenotypes in the inner ear and peripheral nervous system. (**A**) (i–iii) Live images of 4 dpf otic vesicles, lateral view. (i) Wild-type sibling, (ii) *adgrg6^tb233c^*, (iii) *adgrg6^fr24^* showing the swollen, unfused projections in the homozygous mutant otic vesicles in ii and iii compared with the fused pillars in the wild-type ear (marked with dotted lines). (iv–vi) ISH with *vcanb* at 4 dpf. (iv) Wild-type sibling, (v) *adgrg6^tb233c^*, (vi) *adgrg6^fr24^* mutant ears showing overexpression of *vcanb* in the unfused projections. Stronger staining is seen in the stronger allele, *fr24*. (vii–xi) ISH with *mbp* at 4 dpf, (vii-ix) dorsal views, (x–xii) lateral views. (vii, x) wild-type sibling, (viii, xi) *adgrg6^tb233c^*, (ix, xii) *adgrg6^fr24^* showing complex staining patterns in the PNS (black arrows and arrowheads) and CNS (white arrowheads). *mbp* staining around the PLLg is absent in both *tb233c* and *fr24* alleles (black arrowheads); staining in the posterior lateral line nerve is variable in *tb233c* mutants and absent in *fr24* mutants (black arrows). (**B**) Schematic diagram showing the structure of the Adgrg6 receptor and the positions of the predicted amino acid changes for the two *adgrg6* mutant alleles used in this study. (**C**) Light-sheet microscope images using a transgenic line expressing GFP in the otic epithelium, showing a dorsal view of the ear (anterior to the top). (i–iii) Wild-type sibling showing anterior and posterior pillars formed from fused projections (iii). Note that images are flipped horizontally from the originals for ease of comparison (see ***Video 1***; t0 on the stills corresponds to ~100 mins into the video). (iv–vi) Still images from a time-lapse video of *adgrg6^fr24^* mutant with unfused projections that roll around each other (see ***Video 2***). Abbreviations: AC, lumen of anterior semicircular canal; ap, anterior projection; CTF, carboxy-terminal fragment; CUB, Complement C1r/C1s, Uegf, BMP1 domain; ECD, extracellular domain; GAIN, GPCR auto-proteolysis domain; GPS, GPCR proteolytic site; HBD, hormone binding domain; ICD, intracellular domain; NTF, amino-terminal fragment; PC, lumen of posterior semicircular canal; pp, posterior projection; PTX, Pentraxin domain; vp, ventral projection; 7TM, 7-transmembrane domain. Scale bars: 50 µm in Ai, for Aii–vi; 100 µm in Avii, for Aviii–xii; 20 µm in Ci, for Cii–vi.

DOI: https://doi.org/10.7554/eLife.44889.002

The following source data and figure supplements are available for figure 1:

**Figure supplement 1.** Quantification of *mbp* expression around the posterior lateral line ganglion in *adgrg6^tb233c^* mutants and wild-type sibling embryos.

*Figure 1 continued on next page*

*Figure 1 continued*

DOI: https://doi.org/10.7554/eLife.44889.003

**Figure supplement 1—source data 1.** Source data for the percentage area of *mbp* expression shown in *Figure 1—figure supplement 1*.

DOI: https://doi.org/10.7554/eLife.44889.004

screening assay, with the aim of identifying chemical classes capable of rescuing the expression of both genes, which may thus represent agonists of the Adgrg6 signalling pathway. To identify ligands that potentially bind directly to Adgrg6, we then tested hit compounds for their ability to rescue a strong loss-of-function *adgrg6* allele, *fr24*, which predicts a severely truncated protein. Several compounds were unable to rescue *adgrg6*^*fr24* mutants, including a group with similar structures from the gedunin family of compounds. Compounds able to rescue both alleles include colforsin, a known activator of adenylyl cyclase, demonstrating proof-of-principle that our screen can identify compounds that restore GPCR pathway activity downstream of the receptor. These alternative assays for both down-regulation and up-regulation of gene expression, combined with a comparison of rescue in both weak and strong alleles, have facilitated selection of a strong cohort of hit compounds that can be differentiated by the different screens used. Our approach is scalable and can be used to screen additional compound collections. In parallel, chemoinformatics analysis of the compound libraries and identified hits has enabled classification and prioritisation of selected hit compounds.

## Results

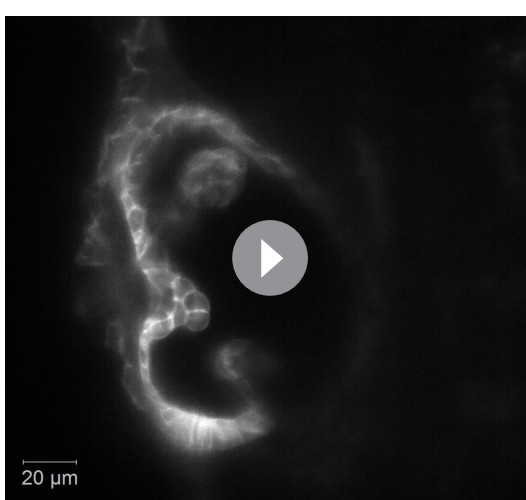

**Video 1.** Light-sheet microscope time-lapse video of the ear shown in *Figure 1Ci-iii*. Dorsal view (anterior to top) of the left inner ear of a phenotypically wild-type sibling embryo showing the anterior, lateral and posterior projections (the anterior projection is partially out of view). In the video, the posterior projection grows and meets the posterior bulge from the lateral projection. The projection and bulge meet, fuse and resolve to form a pillar over 900 min (approximately 55 hpf–70 hpf). The video shows a Maximum Intensity Projection of selected *z*-slices spanning approximately 6 µm, captured every 10 min, and played back at 10 frames per second. Selected stills from the video, flipped horizontally to match the panels showing the mutant ear, are shown in *Figure 1C*.

DOI: https://doi.org/10.7554/eLife.44889.005

### Choice of markers for an in situ hybridisation-based screen: otic *vcanb* expression as a robust readout

We set out to develop a simple assay to identify small molecule modifiers of the Adgrg6 pathway that can be used both to understand Adgrg6 function and to identify compounds that could inform the design of therapeutics. To this end, we chose to perform a drug screen based on in situ hybridisation (ISH), which has the advantage of being a simple, reproducible assay that can be semi-automated (*Baxendale et al., 2012*; *North et al., 2007*). We selected *vcanb* expression in the *adgrg6* mutant ear for our primary screen. *vcanb* has a number of advantages for screening, including highly localised expression in the otic vesicle, very strong and reproducible staining intensity in *adgrg6* mutant embryos, and a clear difference between staining in mutant and wild-type embryos at the stage chosen, making it ideal for manual scoring (*Figure 1A*). We therefore developed a primary screen seeking compounds that can reduce *vcanb* levels in *adgrg6* mutant embryos and rescue the mutant phenotype. We reasoned that, in addition to yielding information for the ear phenotype, compounds that can rescue *vcanb* expression may also rescue myelination defects in the PNS, where expression patterns of genetic markers are more complex and defects are harder to score.

We first made a careful comparison of the otic and PNS defects in weak (*tb233c*) and strong (*fr24*) alleles for the *adgrg6* mutant (*Figure 1A*). The *tb233c* allele is a missense mutation (I963N) in the fourth transmembrane domain of the receptor, whereas the *fr24* allele is a nonsense mutation (L463X), predicting a severely truncated protein lacking the hormone-binding, GAIN, 7TM and C-terminal domains (*Geng et al., 2013*) (*Figure 1B*). Mutants for both *tb233c* and *fr24* alleles have the same defect in semicircular canal formation: otic epithelial projections are enlarged, overgrow, and fail to fuse to form the three pillars that create the hubs of the semicircular canal ducts (*Geng et al., 2013*) (*Figure 1A*). Time-lapse imaging using light-sheet microscopy reveals the dynamics of this process: even when projections make contact with each other, they fail to adhere as in the wild type. Instead, projections in the mutant ear continue to grow, roll around one another as they find space with least resistance, and fill the otic vesicle with semicircular canal projection tissue (*Figure 1C*; *Videos 1* and *2*). In wild-type ears, *vcanb* is expressed in the growing semicircular canal projections between 44 and 72 hr post fertilisation (hpf), but is then strongly down-regulated after fusion; by 4 days post fertilisation (dpf), very little expression is detectable in the ear (*Geng et al., 2013*). By con-

trast, in *adgrg6* mutants, the overgrown and unfused projections in the developing ear continue to express *vcanb* at high levels (*Geng et al., 2013*) (*Figure 1A*). Both alleles show a dramatically increased level of expression over wild-type embryos, but the increase is stronger in the *fr24* allele (*Figure 1A*). mRNA for *adgrg6* itself is expressed in the otic vesicle of mutant embryos for both alleles (*Geng et al., 2013*) (and unpublished data), but it is not known whether a truncated protein including the CUB and PTX domains is produced in the *fr24* allele. (Note, however, that some biological activity is retained for a different truncating mutation, Y782X, in the Adgrg6 NTF [*Petersen et al., 2015*]).

In addition to an upregulation of *vcanb* expression in the ear, the zebrafish *adgrg6* mutant also shows a reduction or loss of expression of the *myelin basic protein* (*mbp*) gene in the PNS (*Geng et al., 2013*; *Monk et al., 2009*). This additional phenotype proved to be very valuable for our screen design, helping to validate hits and eliminate false positives. Expression of *mbp* is present in a complex pattern in wild-type embryos, and shows clear differences between the two alleles, correlating with the predicted severity of the mutations (*Figure 1B*). Expression is variably reduced along the posterior (trunk) lateral line nerve in homozygous mutants for the hypomorphic *tb233c* allele, but in all individuals there is consistent absence of staining in cells (presumed Schwann cells) associated with the posterior lateral line ganglion (PLLg) (*Geng et al., 2013*) (*Figure 1A*, *Figure 1—figure supplement 1*). The *fr24* allele lacks nearly all *mbp* staining along peripheral nerves (*Geng et al., 2013*) (*Figure 1A*). Note that expression of *mbp* in the central nervous system (CNS) is not affected in either allele, obscuring any reduction of *mbp* staining in the PNS without performing a detailed analysis. This

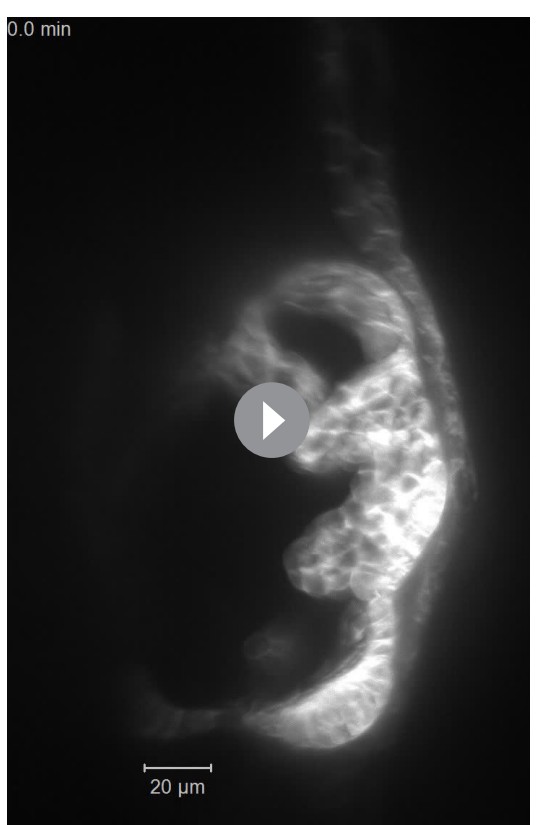

0.0 min

20 μm

**Video 2.** Light-sheet microscope time-lapse video of the ear shown in *Figure 1Civ-vi*. Dorsal view of the right inner ear of an *adgrg6^{fr24}* mutant embryo showing anterior, lateral and posterior projections (the posterior projection is partially out of view). In the video, the anterior projection and anterior bulge from the lateral projection touch, but continue to grow past one another. The unfused projections roll around each other over 900 min (approximately 60 hpf–75 hpf). The video shows a Maximum Intensity Projection of selected *z*-slices spanning approximately 20 μm, captured every 5 min, and played back at 20 frames per second. Selected stills from the video are shown in *Figure 1C*.

DOI: https://doi.org/10.7554/eLife.44889.006

made *mbp* expression unsuitable for a primary screen, but useful for a secondary screen of hit compounds identified from the *vcanb* screen.

## Design of a screening pipeline for compounds that rescue the *adgrg6*[tb233c] mutant phenotype

Our strategy for the screening protocol and analysis pipeline is outlined in *Figure 2*. Both the weak (*tb233c*) and strong (*fr24*) alleles of *adgrg6* mutants are homozygous viable, enabling large batches of 100% mutant embryos to be generated for each assay. We decided to use the hypomorphic allele (*tb233c*) in our primary screen, for four main reasons: (1) adult fish homozygous for the *tb233c* allele produce a larger number of healthy embryos than adults homozygous for the *fr24* allele; (2) a lower concentration of our positive control compound IBMX was sufficient to rescue the phenotype in *tb233c* mutants compared with *fr24* mutants (*Geng et al., 2013*), suggesting that the *tb233c* allele might also be easier to rescue with other compounds in the libraries screened; (3) *vcanb* expression, although not as dramatically affected as in *fr24*, is still robustly over-expressed in the *tb233c* allele, and (4) we predicted that any small molecules that interact with the active site of the receptor or act as allosteric modulators would be missed in a screen using *fr24* mutants, which should only be able to identify compounds acting on targets downstream of the receptor. By using *tb233c*, we should be able to identify modulators of the pathway acting both downstream and at the level of the receptor itself.

## Choice of controls

In all assay plates, we included the phosphodiesterase inhibitor IBMX (100 µM) as a positive control. We have previously shown that addition of 100 µM IBMX at 60 hpf is optimal for both down-regulation of *vcanb* expression and a rescue of projection fusion in the ears of *adgrg6*[tb233c] mutants (*Geng et al., 2013*). At this stage of development, the anterior and posterior projections in the mutant otic vesicle are extended and in close proximity to the lateral projection, to which they would fuse in the wild type (*Figure 2A*). Compounds from both libraries are supplied as stocks dissolved in DMSO; we therefore used 1% DMSO as a negative control. The *nacre* (*mitfa*[-/-]) strain, which has reduced pigmentation, facilitating visualisation of ISH staining patterns, was used as an untreated wild-type control. Three embryos per well were treated with compounds at 25 µM in E3 medium from 60 to 90 hpf, after which they were fixed and analysed for expression of *vcanb* by whole-mount ISH. At the embryonic stage assayed by ISH (90 hpf), expression of *vcanb* in untreated mutant embryos is very specific to the ear, making it clearly visible as two dark spots in the head of each embryo within the well. All controls gave results as expected in all assay plates tested: DMSO-treated mutant embryos showed strong otic staining for *vcanb*, untreated wild-type embryos showed very little staining in the ear, and IBMX-treated mutant embryos showed rescued (down-regulated) otic *vcanb* expression (*Figure 2A*).

## Comparison of compound libraries with diverse structures

In order to test a wide range of compounds, we chose to screen two commercially available small molecule libraries. The Tocriscreen Total library ('Tocris') consists of 1120 compounds representing known bioactive compounds with diverse structures. The Spectrum Collection ('Spectrum'; Microsource Discovery Systems) comprises 2000 compounds, including FDA-approved drugs for repurposing, bioactive compounds and natural products. Scaffold analysis of the two libraries highlights the structural diversity present (*Figure 3—figure supplement 1*). Based on Bemis-Murcko scaffolds (*Bemis and Murcko, 1996*), the Tocris library of 1120 compounds has 693 (62%) scaffolds representing a single compound and only two scaffolds representing more than 10 compounds. The Spectrum library has 682 scaffolds representing unique chemical structures, but as the library consists of 2000 compounds, the proportion of scaffolds represented by a single molecule (30%) is lower than for the Tocris library. Together, the two libraries cover a wide range of chemical space, with a total of 1540 scaffolds, of which 1134 represent unique compounds. Scaffold analysis not only provides a broad overview of the chemical diversity of each library, but can also be used to select and analyse groups of similar compounds with interesting structure-activity relationships. Compounds were also clustered based on their fingerprint similarity using Ward's method of hierarchical agglomerative

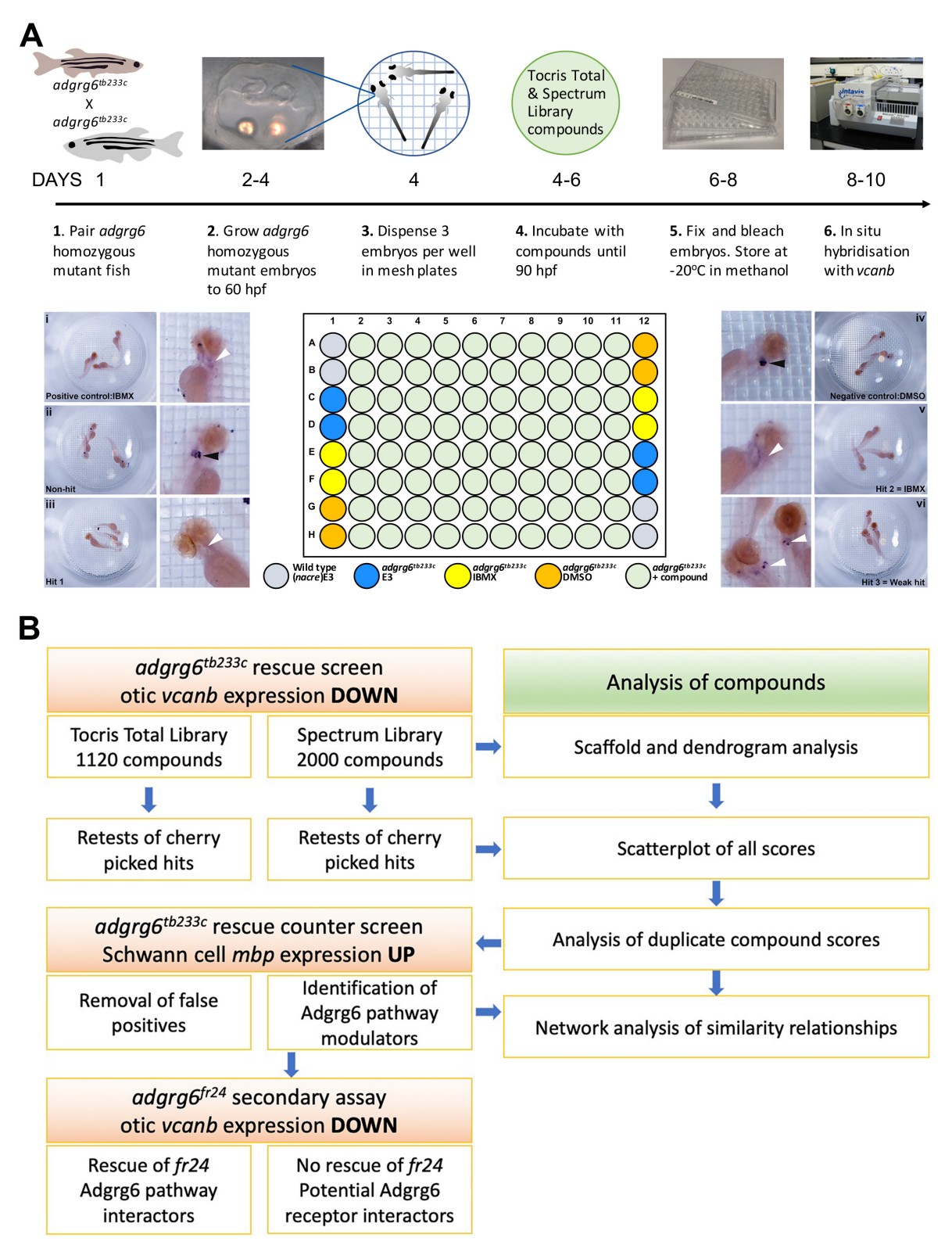

**Figure 2.** Overview of the screening assay protocol and strategy. (**A**) Schematic of the screening assay protocol. Homozygous adult *adgrg6tb233c* mutant fish were paired to raise large numbers of *adgrg6tb233c* mutant embryos. Embryos were grown until 60 hpf, when the lateral, anterior and posterior epithelial projections in the inner ear are evident. Three embryos were aliquoted into each well of a mesh-bottomed multiwell plate in E3 medium. The mesh-bottomed plate was then transferred to the drug plate containing control compounds as shown in the plate layout and library compounds at 25

*Figure 2 continued on next page*

*Figure 2 continued*

µM in 250 µL of E3 embryo medium. Plates were incubated at 28°C until 90 hpf. The mesh-bottomed plate and embryos were then transferred to 4% PFA for fixation (4°C, overnight) and then processed for ISH to *vcanb*. Micrographs show a selection of typical results. Treatment with 100 µM IBMX (positive control, top) results in loss (rescue) of otic *vcanb* expression (white arrowhead). Strong otic *vcanb* expression (black arrowhead) is evident in embryos where the compound had no effect (non-hit) and in negative control wells (not shown). Note the spot of stain in each embryo, marking expression in the otic vesicle. Three examples are shown of wells where compounds were scored as a hit; one of these (Hit 2) was IBMX, represented in the Spectrum collection. (B). Pipeline of the compound screening strategy and chemoinformatics analysis. The left hand side describes the flow of experimental work and the right hand side describes the complementary chemoinformatics processes. For details, see the text.
DOI: https://doi.org/10.7554/eLife.44889.007

clustering, which was useful for visualisation purposes (for dendrograms, see *Figure 3—figure supplement 1*).

## Results of the primary screen for reduction of otic *vcanb* expression levels

To score the efficacy of the compounds in down-regulating *vcanb* mRNA levels, we used a scoring system from 0 to 3 (*Figure 3*; for details, see the Materials and methods). In the primary screen, each compound was tested against three embryos and the score for each embryo was added to give a final score out of 9. The final scores were classified into different groups according to the thresholds shown in *Figure 3B*, with the highest degree of rescue in Category A, representing a combined score no greater than 2. Completion of the primary *vcanb* screen for all 3120 compounds identified 92 (8%) compounds from the Tocris library and 205 (10%) from the Spectrum library that scored in categories A–C (*Figure 3C,E*). 5% of the compounds from each library were found to be either toxic (category F; dead embryos or severe developmental abnormalities) or potentially corrosive (category G; no embryos present), whereas 99 (9%) compounds from Tocris and 269 (13%) from Spectrum were found to cause incomplete or partial suppression of *vcanb* expression (category D). The largest category (E; 2282 compounds from both libraries, 73%), as expected, represented compounds that had no rescuing or other effect at the concentration used (25 µM). To visualise the complete set of screening results and to identify any clusters of hit compounds with similar structures, compounds were displayed as individual data points on a polar scatterplot (*Figure 3D,F*; *Figure 4*; interactive version at https://adlvdl.github.io/visualizations/polar_scatterplot_whitfield_vcanb.html). Compound position along the circumference of the plot for each library is based on position on the respective similarity dendrogram (*Figure 3—figure supplement 1*). Data points that are clustered along radii of the plot are thus more likely to be structurally similar, although note that the juxtaposition of different branches of the dendrogram can also place compounds that differ in structure adjacent to one other. Due to the wide diversity of scaffolds found in the Tocris library, less clustering of hit compounds (A–C) can be observed compared with the molecules in the Spectrum library, where more clusters of compounds in the A–C categories are evident (*Figure 3D,F*).

## Validation of the primary screen: retesting, comparison with control compounds and analysis of duplicates

Possible hit compounds categorised as A–C were selected and arrayed into cherry-pick plates, which were tested using the same assay format. These included all the top hit compounds that scored A or B, and a selection of compounds from the lower-scoring C category (see Materials and methods). Specifically, 83 out of the 92 possible hit compounds from the Tocris library and 145 of the 205 possible hits from the Spectrum library were retested twice, again with three embryos per well. By increasing the number of embryos screened to a total of nine per compound, we aimed to eliminate any false-positive hits that had an increased average score over these two retests or did not show a clear rescue (score >7) in any individual test. In total, 91 compounds from the combined list of hits (29 from Tocris, 62 from Spectrum) that showed consistent rescue of *vcanb* expression across the retests were taken forward for secondary assays (*Figure 4G*).

To provide further validation for the hits identified in the primary screen, we used two approaches. Firstly, we compared the results of our control compounds to those of similar compounds present in the screened compound libraries. The control compound IBMX, a non-selective phosphodiesterase (PDE) inhibitor, is present in the Spectrum library and was identified as a hit in

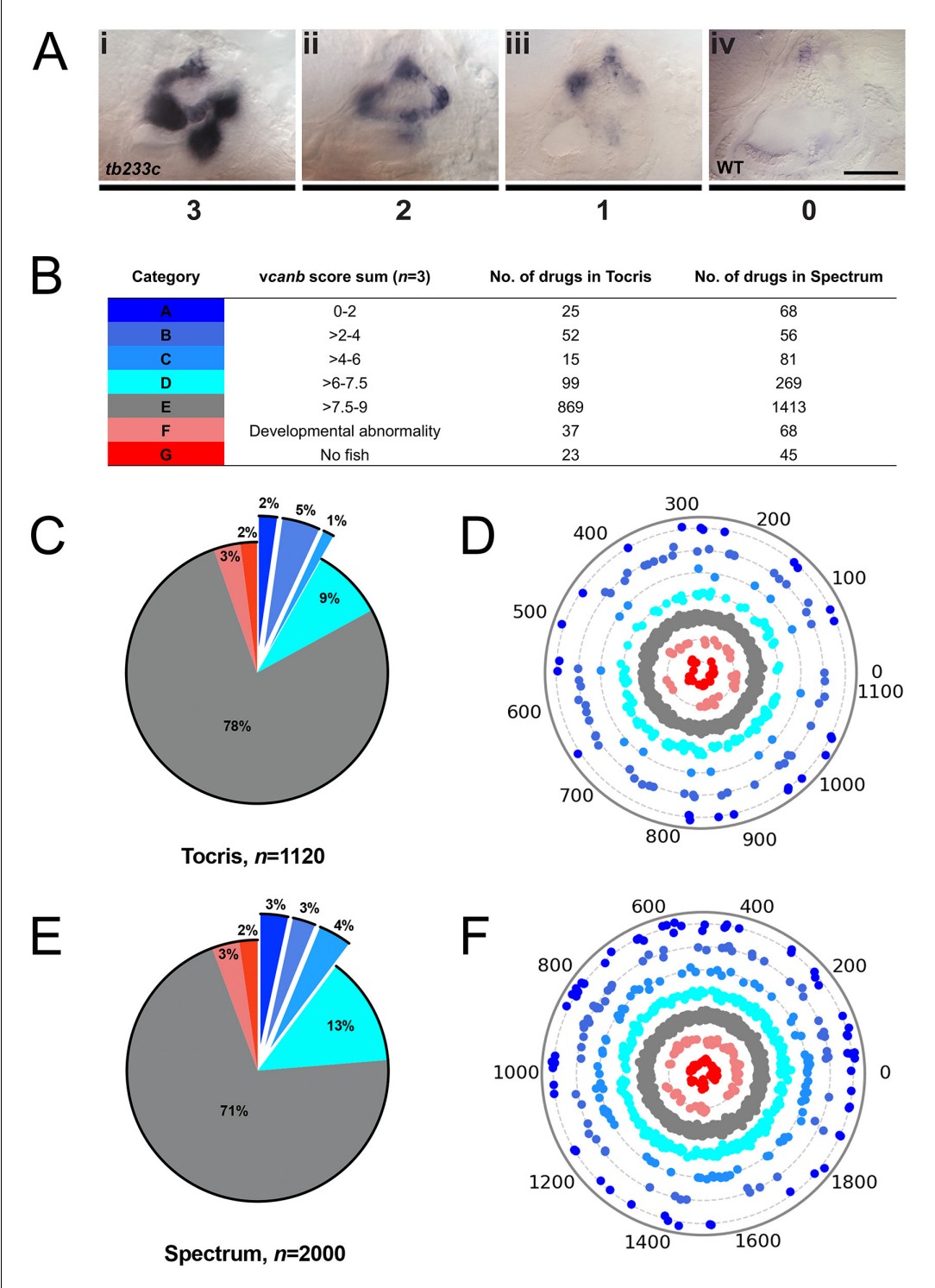

**Figure 3.** A primary drug screen identified 92 (Tocris) and 205 (Spectrum) putative hit compounds able to down-regulate *vcanb* mRNA expression in *adgrg6*<sup>tb233c</sup> mutants. (A) Scoring system used to assess *vcanb* mRNA expression levels in the inner ear of *adgrg6*<sup>tb233c</sup> embryos after treatment. (Ai) *vcanb* mRNA expression in the untreated/DMSO-treated *adgrg6*<sup>tb233c</sup> mutant ear (score 3). Scores 2 (Aii) and 1 (Aiii) were given to embryos that showed reduced *vcanb* mRNA expression to some extent, with 1 given for a stronger down-regulation than 2. (Aiv) Score 0 was given to embryos where *vcanb* mRNA levels were equivalent to wild-type levels. (B) Compounds were categorised A–G according to the total *vcanb* score from the three embryos treated. Colours for each category correspond to the colours used in panels C–F. (C, E) Pie charts showing the distribution of compounds from the Tocris (C) and Spectrum (E) libraries in categories A–G. (D, F) Compounds from the Tocris and Spectrum libraries were ordered according to

*Figure 3 continued on next page*

*Figure 3 continued*

similarities in their chemical structure and presented as individual dots in polar scatterplots in D and F, respectively, with jitter (noise) introduced to improve visualisation. The Spectrum library results have a higher level of clustering as expected from the scaffold analyses. Scale bar: 50 µm.

DOI: https://doi.org/10.7554/eLife.44889.008

The following source data and figure supplement are available for figure 3:

**Source data 1.** Source data for *Figure 3D*.
DOI: https://doi.org/10.7554/eLife.44889.011
**Source data 2.** Source data for *Figure 3F*.
DOI: https://doi.org/10.7554/eLife.44889.012
**Figure supplement 1.** Scaffold analysis of compound structures in the Tocriscreen Total and Spectrum libraries.
DOI: https://doi.org/10.7554/eLife.44889.009

the primary screen (*Figure 2A*). The most similar compound to IBMX from both libraries is 8-methoxymethyl-3-isobutyl-1-methylxanthine (MMPX), a selective PDE-1 inhibitor. MMPX is present in the Tocris library, but was not identified as a hit in our screen, most likely due to its selectivity. In previous work we also used forskolin to raise cAMP levels and rescue the *adgrg6* ear phenotype (*Geng et al., 2013*), but forskolin requires different assay conditions with short drug incubation times to avoid toxicity. Forskolin is represented in the Tocris library, but was toxic in our screening assay. The Spectrum Collection contains two forskolin-related compounds, colforsin and desacetylcolforsin. Colforsin, a water-soluble derivative of forskolin, was identified as a hit in the primary screen, and retested positive in all subsequent tests (see also below); it appeared to be less toxic than forskolin, whereas desacetylcolforsin was toxic at the concentration used. The identification of both IBMX and colforsin as hits in the primary screen confirmed that the assay conditions used were efficient at detecting expected hit compounds.

Secondly, we compared the scores for all compounds that were duplicated in both compound libraries. Chemoinformatics analysis of the Tocris and Spectrum libraries identified 155 compounds represented in both libraries, 65% of which (100/155) had exactly the same *vcanb* score average from the two individual screens. 39 (25%) of the 155 duplicate compounds yielded a *vcanb* score average that differed by 1–2 units between the two libraries; 12 (8%) of the compounds yielded a *vcanb* score average that differed by 3–6 units, whereas only 4 (3%) compounds had a score average that differed by 7–9 units. In summary, 90% (139/155) of the compounds common to both libraries showed similar scores for the *vcanb* assay from each library (scores differing by ≤2 units), whereas 10% (16/155) of the compounds resulted in differing levels of *vcanb* down-regulation between the two different libraries. After retesting, the difference between the two *vcanb* score averages for nine of these compounds was reduced; however, for seven compounds, the scores between the two libraries remained significantly different. These discrepancies could be either due to differences in compound purity between the two suppliers, or could be due to experimental error (e.g. in the concentration used, or during the ISH protocol). In cases where the same compound was scored as toxic in one assay and not in another, the health condition of the embryos in a particular well could be the underlying reason. Duplicated compounds have been included in the data for each library in the polar scatter plots (*Figures 3* and *4*).

The top 91 hit compounds from both libraries (29 from Tocris, 62 from Spectrum) that scored A–C in all three *vcanb* assays were combined to give a complete list of 89 unique compounds, with baicalein and gedunin present in both libraries. The list covers a wide spectrum of naturally-derived and synthetic molecules, with known and unknown functions (*Supplementary file 1*). The hit compounds with known functions include calcium channel blockers, antifungal, anti-inflammatory, antihyperlipidemic, antibacterial and anthelmintic agents, as well as compounds with known antineoplastic and vasodilatory properties.

## Secondary screen for rescue of *mbp* expression, and identification of false positives

The two retests for *vcanb* expression significantly reduced the possibility of false-positive results due to experimental error (e.g. in the ISH protocol), but the list of hits could still contain false-positive compounds that may generally inhibit transcription. In order to eliminate such compounds, we exploited the expression of *mbp* as a secondary screening assay, scoring for rescue of expression

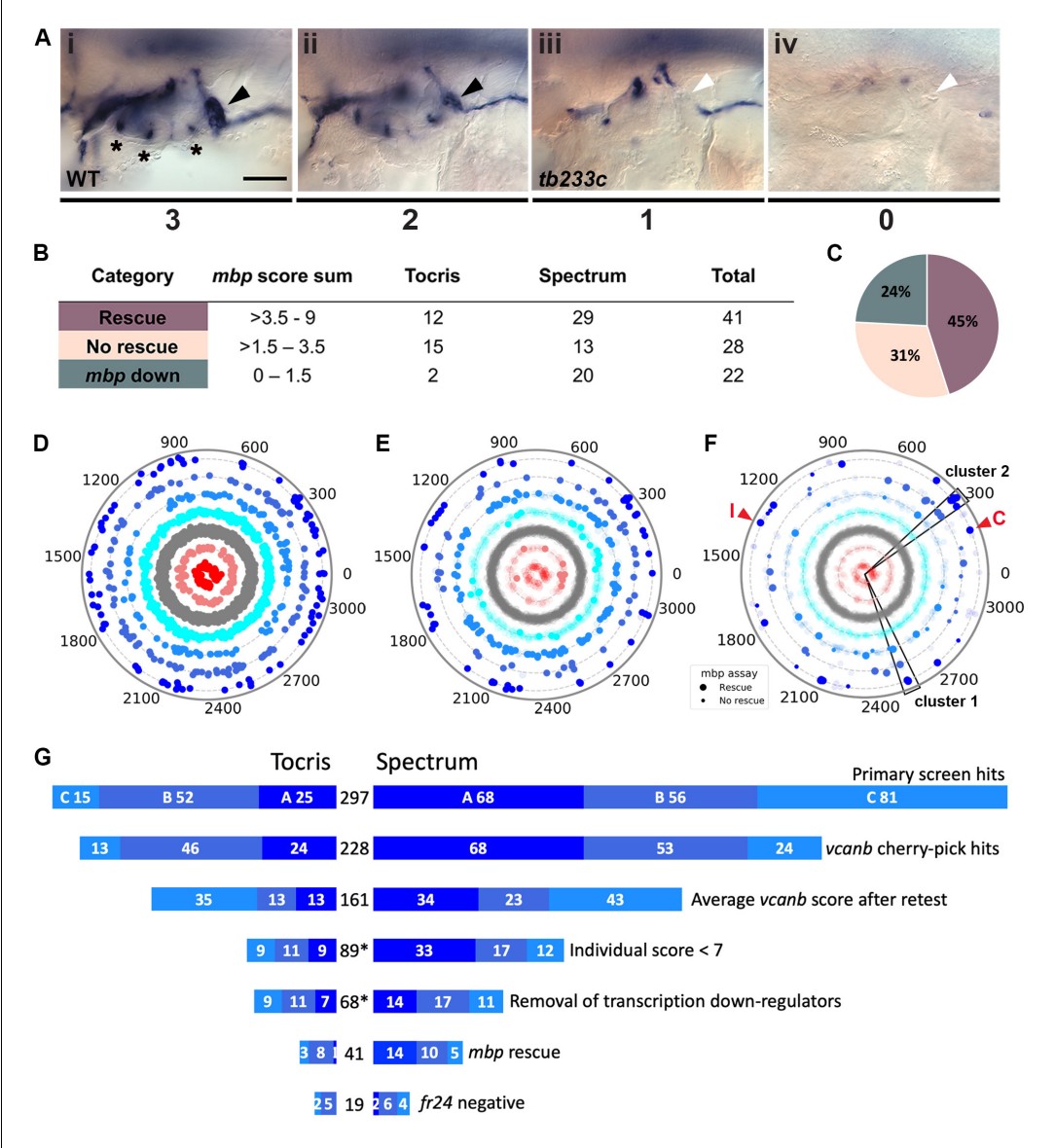

**Figure 4.** Retesting and counter screen for *mbp* expression reveals chemical clustering of hit compounds. (A) Scoring system used to assess *mbp* mRNA expression levels around the PLLg of *adgrg6*^*tb233c*^ embryos after treatment. (Ai) A score of 3 was given to embryos where *mbp* mRNA expression was similar to wild-type levels. Black arrowhead: *mbp* expression around the PLLg. (Aii) A score of 2 was given to embryos that showed weak *mbp* expression around the PLLg. (Aiii) A score of 1 was given to embryos with *mbp* expression identical to that in untreated *adgrg6*^*tb233c*^ mutants (absence of *mbp* expression around the PLLg (white arrowhead), with weak expression elsewhere). (Aiv) A score of 0 was used to indicate embryos where *mbp* mRNA expression was absent throughout the PNS. Asterisks mark expression near the three cristae of the ear. Scale bar: 50 μm. (B, C) *mbp* scoring system and classification of the compounds. (B) Compounds were categorised according to the *mbp* score sum from three embryos (average from two experiments; six embryos total) and grouped into compounds able to rescue *mbp* expression (score >3.5–9) and unable to rescue *mbp* expression (>1.5–3.5). A third class of compounds down-regulated both *vcanb* and *mbp* (score 0–1.5) and were not followed further. (C) Distribution of the compounds in the different rescue categories after the *mbp* counter screen. (D) Compounds from both libraries are represented as individual dots in a combined polar scatterplot (3120 compounds in total; https://adlvdl.github.io/visualizations/polar_scatterplot_whitfield_vcanb.html). Compounds were ordered according to similarities in their chemical structure and placed in concentric circles according to the category A–G they were assigned to after the primary screen, with jitter (noise) introduced to improve visualisation. (E) Polar scatter plot of the 91 hit compounds that passed the first retest and were followed up with *mbp* counter screens; previous scores for the compounds not followed are faded. (F) Polar scatter plot of the final 68 hit compounds (non-faded) after *mbp* counter screens. Bigger dots represent compounds that rescued *mbp* expression, whereas smaller dots correspond to the compounds that did not rescue *mbp* expression; compounds that downregulated *mbp* expression, or were not followed, are faded. Wedges on the scatter plot delineate the two clusters of compounds with similar structures for which some hits were followed up in further analysis (see text). The positions of IBMX (I) and colforsin (C) are indicated (red arrowheads). (G) Overview of the hit selection process. The length of the horizontal bars is

*Figure 4 continued on next page*

*Figure 4 continued*

proportional to the number of hit compounds taken through to each stage. Data for the Tocris library are on the left-hand side; data for the Spectrum library are on the right-hand side. The proportion of compounds in hit categories A, B and C are shown using the same colour scheme as in *Figure 3*, with the top bar representing the number of hits from the primary screen listed in *Figure 3B*. The second bar shows the number of compounds that were cherry-picked. The average scores from nine embryos (after retests) is shown in the third bar. Note that some compounds will change category after the retests and the number of category C compounds is increased. Any compounds that failed to rescue in any single retest were also not taken forward (fourth bar). The *mbp* data (E) are represented in the fifth and sixth bars. The final bar represents the compounds that were unable to rescue the strong *fr24* allele. The total number of compounds at each stage is shown in the centre. Asterisks denote numbers that do not include duplicate compounds.

DOI: https://doi.org/10.7554/eLife.44889.013

The following source data is available for figure 4:

**Source data 1.** Source data for *Figure 4D*.
DOI: https://doi.org/10.7554/eLife.44889.014

around the posterior lateral line ganglion (PLLg) (*Figure 4A*; Materials and methods). This counter screen has the advantage of assessing for up-regulation (restoration) of *mbp* expression in mutant embryos, in contrast to the down-regulation of *vcanb* expression in the primary screen. All compounds that passed the first retest for *vcanb* (89 compounds in total) were subjected to this secondary assay for *mbp* expression. We used the same assay format and treatment time window as for *vcanb*, as we had previously found that treatment with IBMX between 60 and 90 hpf was also able to rescue *mbp* expression in *adgrg6* mutants (not shown).

Following two experimental repeats ($n$ = 6 fish tested per drug), compounds were categorised into groups based on their average *mbp* score. These included groups of compounds that showed rescue of the mutant phenotype (an increase of *mbp* expression, specifically around the PLLg); no rescue (*mbp* expression equivalent to that in untreated *adgrg6^{tb233c}* mutants), and those that resulted in a decrease in *mbp* expression, as shown in *Figure 4A*. We identified 41 compounds (12 from Tocris, 29 from Spectrum) that rescued *mbp* expression and thus represent possible modulators of Adgrg6 pathway (*Figure 4B,C*; *Table 1*). Twenty-eight hit compounds (15 from Tocris, 13 from Spectrum) strongly down-regulated *vcanb* expression but did not affect *mbp* expression in *adgrg6^{tb233c}* mutants. These could represent compounds that can rescue *vcanb* expression in an inner ear-specific or Adgrg6-independent manner. Alternatively, as all the assays were carried out at a single concentration (25 µM), it is possible that some or all of these compounds could rescue *mbp* expression at a higher concentration (as is the case for IBMX with the *fr24* allele). The 28 members of this group are structurally and functionally diverse (*Supplementary file 1*). Finally, 22 compounds (two from Tocris, 20 from Spectrum) reduced the expression of both *vcanb* and *mbp*. This latter group—potential false positives in the *vcanb* assay—could represent general inhibitors of transcription or development, and were excluded from further analysis, resulting in a final list of 68 hit compounds (*Supplementary file 1*). The heat map in *Figure 5A* displays these groups using data from each of the screens and retests and clusters the compounds based on their activity.

## Compounds that can rescue both inner ear and myelination defects

The 41 compounds that could both down-regulate *vcanb* expression and restore *mbp* expression to wild-type levels in *adgrg6^{tb233c}* mutants, presumed modulators of the Adgrg6 signalling pathway (*Table 1*), are highlighted on the final combined polar scatter plot (*Figure 4F*). Although hit compounds are scattered around the plot, some clustering is evident, and we chose two groups for further analysis (*Figure 4F*; boxes at 300, 2500); these clusters are also clearly seen in a compound network display based on structural similarity in *Figure 5B*; interactive version at https://adlvdl. github.io/visualizations/network_whitfield_vcanb_mbp/index.html). Groups with five or more compounds included the pyridines (cluster 1, *Figures 4D* and *5B*) and the tetranortriterpenoids (gedunin derivatives) (cluster 2, *Figures 4D* and *5B*). The pyridine cluster included one pyrazolopyridine and six dihydropyridines, a class of L-type calcium channel blockers with vasodilatory properties (reviewed in *Tocci et al., 2018*). The gedunins are a family of naturally occurring compounds, previously attributed with antineoplastic and neuroprotective effects (*Jang et al., 2010*; *Subramani et al., 2017*).

**Table 1.** List of the 41 hit compounds that rescued the expression of both *vcanb* and *mbp* in *adgrg6*[tb233c] mutants, thus representing putative Adgrg6 pathway modulators.

The table includes the plate and well ID, along with known activities and the average score from nine *adgrg6*[tb233c] embryos in the *vcanb* assay, from six *adgrg6*[tb233c] embryos in the *mbp* assay and from three *adgrg6*[fr24] embryos in the *fr24* assay. Grey shading indicates compounds presumed to interact with Adgrg6 receptor directly; yellow shading indicates compounds presumed to be downstream effectors of the pathway. Abbreviations: DE, dead embryos; ND, no data; S, Spectrum; T, Tocris. *Note that cilnidipine can rescue *fr24* at 40 µM (data not shown). See also **Table 1—source data 1**.

| # | Plate | Well | Compound name | Known activity | *vcanb* score | *mbp* score | *fr24* score |
|---|-------|------|---------------|----------------|---------------|-------------|--------------|
| 1 | S18 | C09 | CARAPIN-8(9)-ENE | undetermined | 0.00 | 8.50 | 9.00 |
| 2 | S25 | D08 | 3-ISOBUTYL-1-METHYLXANTHINE (IBMX) | phosphodiesterase inhibitor, non-selective adenosine receptor antagonist | 2.00 | 8.50 | 9.00 |
| 3 | S17 | F05 | DEOXYGEDUNIN | neuroprotective | 2.00 | 8.00 | 9.00 |
| 4 | S23 | F10 | DIHYDROFISSINOLIDE | undetermined | 2.67 | 7.50 | 9.00 |
| 5 | S04 | B02 | IVERMECTIN | antiparasitic | 2.33 | 7.00 | 9.00 |
| 6 | T01 | F06 | SC-10 | protein kinase C activator, NMDA receptor activator | 5.67 | 6.50 | 9.00 |
| 7 | T01 | H11 | 1,3-Dipropyl-8-phenylxanthine | Selective adenosine A1 receptor antagonist | 3.33 | 6.50 | 9.00 |
| 8 | S17 | E02 | 3-DEOXO-3beta-ACETOXYDEOXYDIHYDROGEDUNIN | undetermined | 0.00 | 6.50 | 9.00 |
| 9 | T11 | F07 | Cilnidipine* | dihydropyridine N- and L-type $Ca^{2+}$ channel blocker | 2.00 | 6.50 | 9.00 |
| 10 | S13 | F03 | AMIODARONE HYDROCHLORIDE | coronary vasodilator, $Ca^{2+}$ channel blocker | 5.00 | 6.50 | 9.00 |
| 11 | S06 | E02 | HYDROCORTISONE HEMISUCCINATE | glucocorticoid | 3.67 | 6.00 | 9.00 |
| 12 | T01 | C04 | (RS)-(Tetrazol-5-yl)glycine | highly potent NMDA receptor agonist | 3.00 | 5.00 | 9.00 |
| 13 | S02 | E05 | LOMEFLOXACIN HYDROCHLORIDE | antibacterial | 5.33 | 5.00 | 9.00 |
| 14 | S13 | E04 | ETHAMIVAN | CNS & respiratory stimulant | 4.67 | 5.00 | 9.00 |
| 15 | T08 | B04 | CGS 15943 | potent adenosine receptor antagonist | 5.33 | 4.50 | 9.00 |
| 16 | S13 | E09 | ASTEMIZOLE | H1 antihistamine (nonsedating) | 4.67 | 4.50 | 9.00 |
| 17 | T02 | A09 | SKF 91488 dihydrochloride | histamine N-methyltransferase inhibitor | 3.00 | 4.00 | 9.00 |
| 18 | S25 | F05 | 11alpha-HYDROXYPROGESTERONE HEMISUCCINATE | glucocorticoid | 2.67 | 4.00 | 9.00 |
| 19 | T14 | A07 | Efonidipine hydrochloride monoethanolate | dihydropyridine L-type and T-type $Ca^{2+}$ channel blocker | 3.67 | 4.00 | 9.00 |
| 20 | T05 | C09 | Nifedipine | dihydropyridine L-type $Ca^{2+}$ channel blocker | 4.33 | 7.00 | 8.00 |
| 21 | T05 | E08 | CGP 37157 | antagonist of mitochondrial $Na^+/Ca^{2+}$ exchange | 3.67 | 6.50 | 8.00 |
| 22 | S05 | D03 | DANAZOL | anterior pituitary suppressant, anti-estrogenic | 1.00 | 5.00 | 8.00 |
| 23 | S18 | H09 | XANTHYLETIN | undetermined | 1.00 | 4.50 | 8.00 |
| 24 | S18 | A06 | FERULIC ACID | antineoplastic, choleretic, food preservative | 3.67 | 4.00 | 8.00 |
| 25 | S18 | F02 | alpha-DIHYDROGEDUNOL | undetermined | 2.33 | 4.00 | 8.00 |
| 26 | T05 | F04 | (S)-(+)-Niguldipine hydrochloride | dihydropyridine L-type $Ca^{2+}$ channel blocker, α1 antagonist | 3.67 | 5.00 | 7.00 |
| 27 | T07 | F02 | Tracazolate hydrochloride | subtype-selective $GABA_A$ allosteric modulator | 2.33 | 4.50 | 7.00 |
| 28 | S10 | E02 | NIMODIPINE | dihydropyridine L-type $Ca^{2+}$ channel blocker | 0.33 | 7.00 | 6.00 |

*Table 1 continued on next page*

*Table 1 continued*

| # | Plate | Well | Compound name | Known activity | *vcanb* score | *mbp* score | *fr24* score |
|---|-------|------|---------------|----------------|---------------|-------------|--------------|
| 29 | S17 | E06 | 3beta-ACETOXYDEOXODIHYDROGEDUNIN | undetermined | 2.00 | 4.50 | 5.00 |
| 30 | S17 | F02 | DIHYDROGEDUNIN | undetermined | 1.67 | 5.00 | 2.00 |
| 31 | S22 | F09 | TANGERITIN | undetermined | 1.33 | 5.50 | 1.00 |
| 32 | S10 | F07 | COLFORSIN | adenylate cyclase activator, antiglaucoma, hypotensive, vasodilator | 0.00 | 9.00 | 0.00 |
| 33 | T04 | G02 | Imiloxan hydrochloride | selective $\alpha_{2B}$-adrenoceptor antagonist | 0.67 | 9.00 | ND |
| 34 | S24 | C03 | 3alpha-ACETOXYDIHYDRODEOXYGEDUNIN | undetermined | 0.33 | 8.50 | DE |
| 35 | S11 | E02 | EZETIMIBE | antihyperlipidemic (sterol absorption inhibitor) | 2.00 | 7.50 | 0.00 |
| 36 | S10 | E06 | NITRENDIPINE | dihydropyridine L-type $Ca^{2+}$ channel blocker | 1.33 | 7.00 | ND |
| 37 | S11 | E08 | ROSUVASTATIN CALCIUM | antihyperlipidemic | 0.00 | 6.00 | 0.00 |
| 38 | S22 | C07 | DEMETHYLNOBILETIN | undetermined | 0.00 | 6.00 | 0.00 |
| 39 | S22 | G11 | HEXAMETHYLQUERCETAGETIN | undetermined | 0.00 | 5.50 | DE |
| 40 | S22 | F08 | NOBILETIN | matrix metaloproteinase inhibitor, antineoplastic, anti-ERK, NF-κB suppressor | 0.00 | 5.00 | DE |
| 41 | S12 | H07 | PREGNENOLONE SUCCINATE | glucocortcoid, antiinflammatory | 4.67 | 4.00 | DE |

DOI: https://doi.org/10.7554/eLife.44889.016

The following source data is available for Table 1:

Source data 1. Source data for *Table 1*.

DOI: https://doi.org/10.7554/eLife.44889.017

A selection of compounds was chosen for further study (*Figures 6–8*). Two dihydropyridines, nifedipine and cilnidipine, were chosen from cluster 1. The third compound chosen was tracazolate hydrochloride, a pyrazolopyridine derivative belonging to the nonbenzodiazepines and a known γ-aminobutyric acid A (GABAA) modulator (*Thompson et al., 2002*), which strongly down-regulated *vcanb* expression to wild-type levels. FPL 64176 was also chosen for further analysis, based on its potent efficacy in down-regulating *vcanb,* and the fact that it was the only calcium channel modulator (*Liu et al., 2003*) that did not rescue *mbp* expression efficiently. Initial experiments to repeat the rescue of the *vcanb* and *mbp* expression with freshly-sourced compounds from alternative suppliers (see Materials and methods) confirmed that the pyridines cilnidipine, nifedipine and tracazolate hydrochloride were able to decrease otic *vcanb* expression and increase *mbp* expression around the PLLg in mutant embryos for the *tb233c* allele, whereas FPL 64176 was able to reduce *vcanb* expression but was unable to restore *mbp* expression to wild-type levels (*Figure 6C*).

To examine whether the network clustering was able to predict functional activity, we selected an additional four dihydropyridines that were not represented in the Tocris or Spectrum libraries, and tested whether they could also rescue *vcanb* expression in *adgrg6tb233c* mutants. Nilvadipine, nemadipine-A, felodipine and lercanidipine are all dihydropyridine calcium channel blockers of the type used to treat hypertension. Nilvadipine is structurally closely related to nifedipine (*Figure 6B*); as predicted, it gave a dose-responsive rescue of otic *vcanb*, with full rescue at 50.6 μM (*Figure 6Cvii*, *Figure 7—figure supplement 1*), and a strong rescue of *mbp* expression at 22.5 μM (*Figure 6Cxiv*). The three other compounds showed a range of efficacy in the *vcanb* assay at 25 μM, with nemadipine-A showing complete rescue, felodipine mild rescue and lercanidipine no rescue (*Figure 7—figure supplement 1*). However, a higher concentration of lercanidipine (50 μM) was able to rescue *vcanb* expression, whereas felodipine continued to show mild rescue at 40 μM (*Figure 7—figure supplement 1*).

We also tested a higher concentration of a Tocris compound from the pyridine cluster, (±)-Bay K 8644, which had originally scored as a non-hit in the primary screen (25 μM). We found that this compound rescued otic *vcanb* expression effectively at 40 μM, and in fact also gave a mild rescue at 25

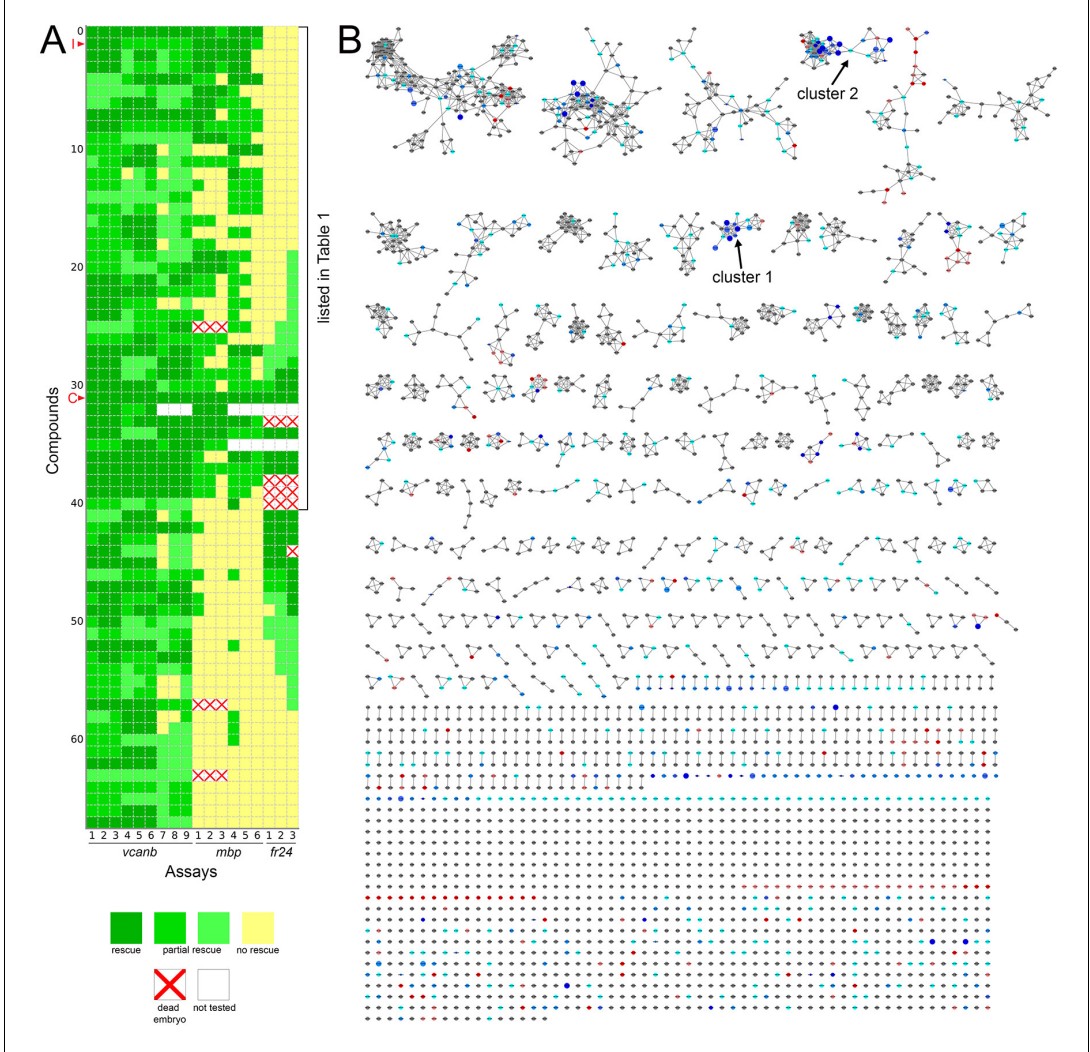

**Figure 5.** Heatmap of the assay results and network analysis for 68 compounds identified in the *vcanb* screen. (**A**) Heatmap of the assay results for each of the 68 hit compounds. Each box represents an embryo screened in each of the three assays (*vcanb*, *mbp* and *fr24*) as listed at the bottom of the heatmap. Each row corresponds to a different compound. Colours correspond to the scoring system used for each screen (0–3), with dark green, a strong hit (rescue of the mutant phenotype); yellow, no rescue; white, no data; white with red cross, toxic. Compounds were sorted based on the average score for *mbp* with strongest rescue at the top. The bracket indicates the 41 compounds that rescued both *vcanb* and *mbp* expression in *adgrg6*^*tb233c*^ mutants and thus represent putative Adgrg6 pathway modulators. Abbreviations: C, colforsin; I, IBMX. (**B**) Network analysis based on structural similarity, showing all 3120 compounds from the two libraries. Compounds that rescued *mbp* expression are shown as larger nodes; compounds that did not rescue *mbp* expression are shown as smaller nodes. The colours used for compounds/nodes correspond to categories A–G (as indicated in *Figure 3*) and the two clusters of structurally similar compounds highlighted in *Figure 4* are also shown here. An interactive version of this figure can be accessed and mined at: https://adlvdl.github.io/visualizations/network_whitfield_vcanb_mbp/index.html.
DOI: https://doi.org/10.7554/eLife.44889.015

µM in this experiment (*Figure 7—figure supplement 1*). Interestingly, (±)-Bay K 8644 is a dihydro-pyridine that acts as a calcium channel agonist with similar activity to FPL 64176 (*Hu et al., 2013*; *Rampe et al., 1993*). All compounds tested from cluster one share a large common substructure composed of a pyridine ring and its five substitutions (one phenyl ring, two ester groups, and two methyl groups). Felodipine and nemadipine-A both have several halogen atoms bound to the phenyl ring, whereas on most of the other compounds a nitro group is bound to the phenyl ring. The main difference between the dihydropyridine structures comes from the variety of different esters attached to the pyridine ring (*Figure 6C*, *Figure 7—figure supplement 1B*).

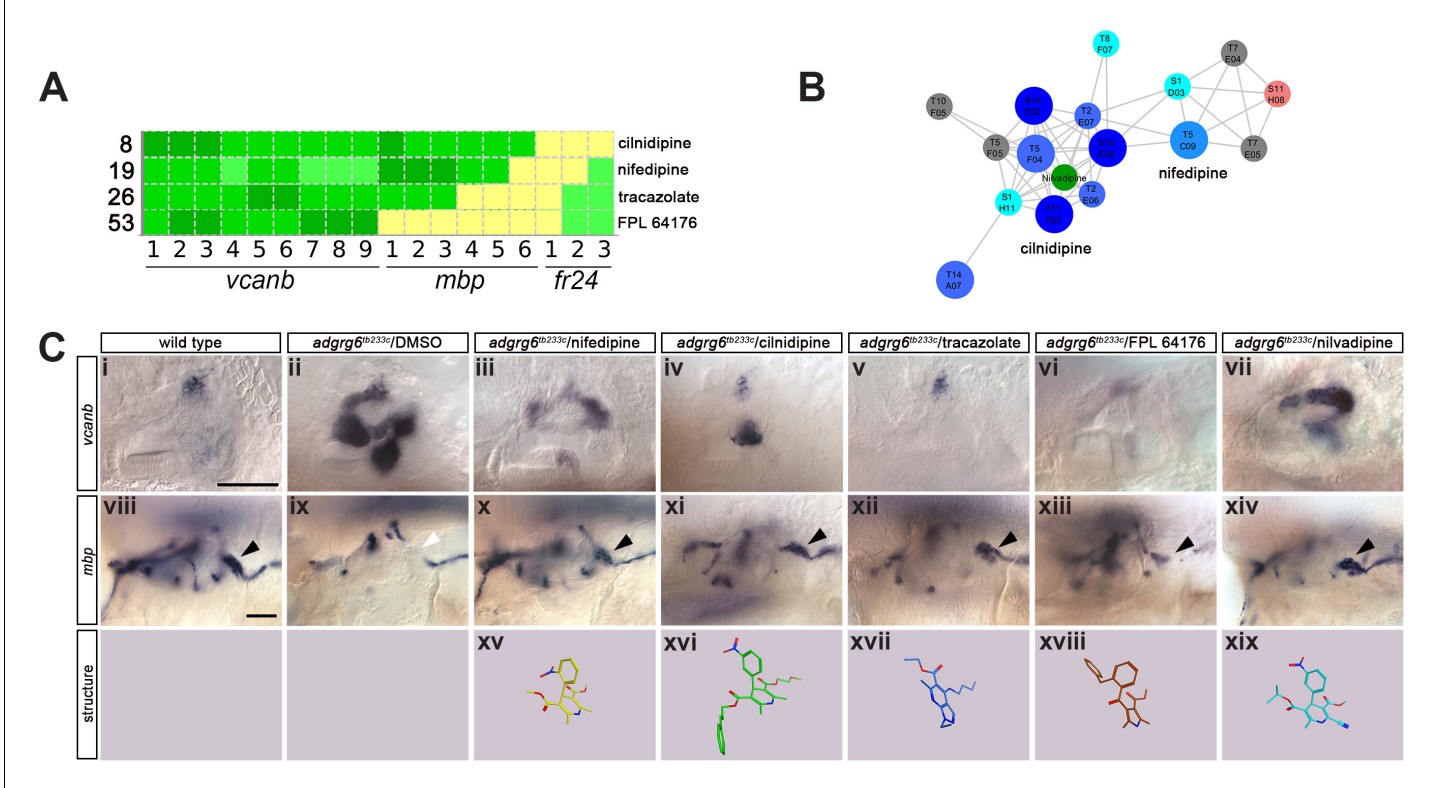

**Figure 6.** Hit compounds from the *vcanb* screen vary in their ability to restore *mbp* expression in *adgrg6*[tb233c] mutant embryos. (**A**) Section of the heatmap in *Figure 5A* showing the results for nifedipine, cilnidipine, tracazolate hydrochloride and FPL 64176. (**B**) Enlargement of the dihydropyridine cluster (cluster 1 in *Figures 4G* and *5B*), highlighting cilnidipine and nifedipine. Compounds that rescued *mbp* expression are shown as larger nodes, whereas compounds that did not rescue *mbp* expression are shown as smaller nodes. The relationship of nilvadipine (green circle) to the other compounds in this cluster is also illustrated. (**C**) (i–vii) Lateral images of the inner ear at 4 dpf stained for *vcanb* by ISH. (i) Wild-type, (ii) *adgrg6*[tb233c] mutant treated with DMSO as a control, (iii–vii) treatment of *adgrg6*[tb233c] mutants with test compounds at 25 μM, with the exception of nilvadipine, which was tested at 22.5 μM. (viii–xiv) *mbp* mRNA expression of embryos treated as indicated above. Black arrowheads indicate *mbp* expression around the PLLg; white arrowhead in (ix) indicates the position of the PLLg in the untreated mutant, lacking *mbp* expression. Nifedipine, cilnidipine, tracazolate hydrochloride and nilvadipine all rescued *mbp* expression around the PLLg, whereas FPL 64176 did not rescue *mbp* expression around the PLLg so efficiently. (xv–xix) Representation of the chemical structures of the five compounds tested. Scale bar in (i), 50 μm (applies to i-vii); scale bar in viii, 50 μm (applies to viii–xiv).

DOI: https://doi.org/10.7554/eLife.44889.018

## Nifedipine, cilnidipine, tracazolate hydrochloride and FPL 64176 rescue otic defects in *adgrg6*[tb233c] mutants in a dose-dependent manner

The four compounds shown in *Figure 6* were also selected for dose-response assessment, by exposing *adgrg6*[tb233c] embryos to concentrations ranging from 0.3 μM to 222.2 μM between 60–110 hpf. Nine embryos were tested for each concentration, and a 1.5-fold dilution series of each drug was used. ISH analysis of the 110-hpf embryos revealed a robust, dose-dependent down-regulation of *vcanb* mRNA expression in response to treatment with all four drugs (*Figure 7*). Expression of *vcanb* mRNA was assessed by annotating each embryo with two scores, one representing the number of unfused projections stained (*Figure 7A*), and the other representing the intensity of the stain (*Figure 7B*, score as in *Figure 3A*). All four drugs were able to reduce both the intensity of the ISH staining and the number of projections stained in the ear in a dose-dependent manner. For each of the four drugs, the intensity of the *vcanb* staining decreased even after treatment with low doses, whereas higher doses were needed to reduce the number of the projections stained.

In order to investigate whether other aspects of the ear phenotype in *adgrg6*[tb233c] mutants could be rescued by compound treatment, the inner ears of live treated embryos were observed with differential interference contrast (DIC) optics at 110 hpf (or 90 hpf in the case of FPL 64176, due to its

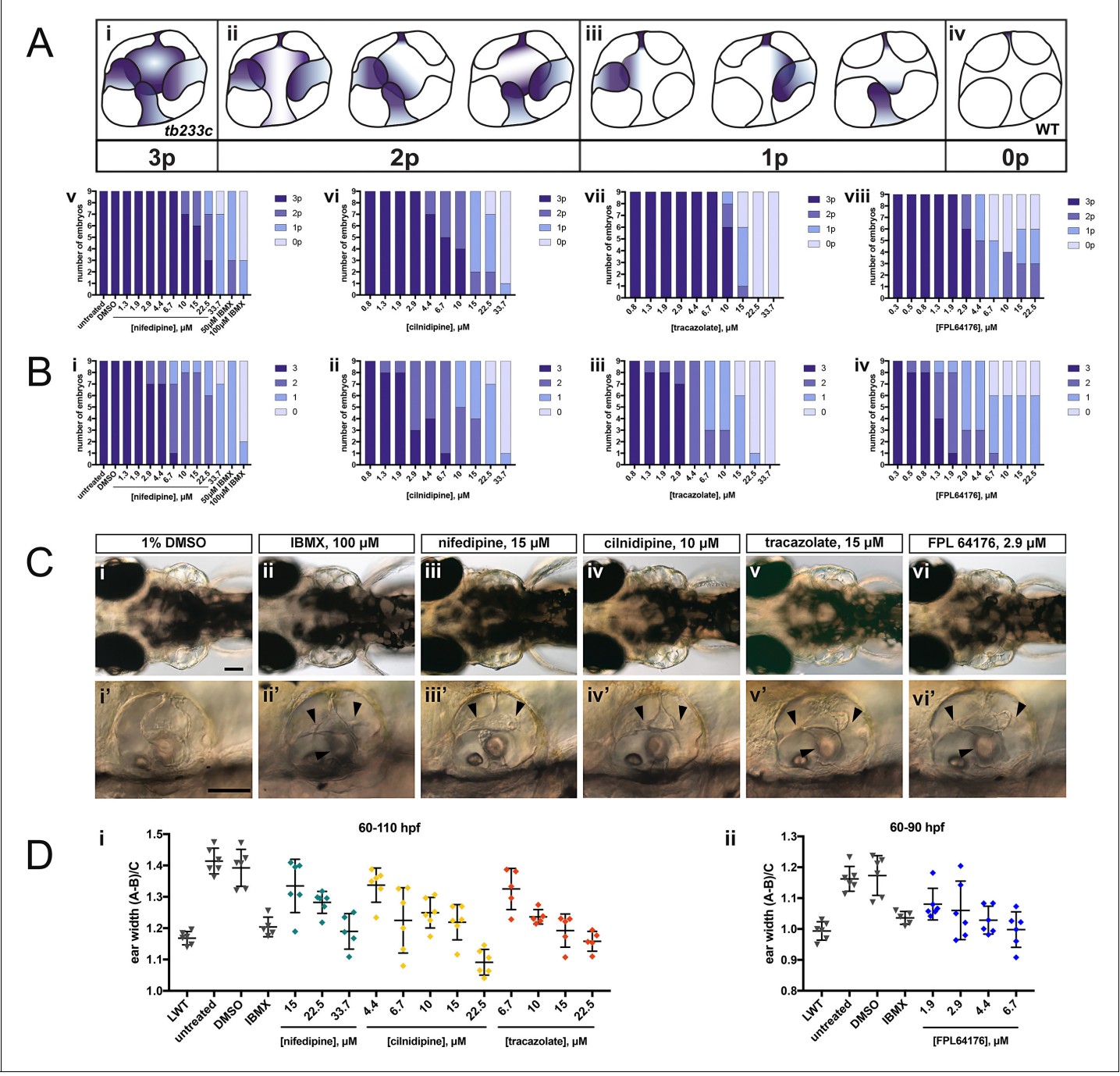

**Figure 7.** Selected hit compounds rescue the *adgrg6*tb233c mutant ear phenotype in a dose-dependent manner. *adgrg6*tb233c homozygous embryos were exposed to a 1.5-fold dilution series of concentrations (ranging from 0.3 µM to 33.7 µM), tailored to the toxicity of nifedipine, cilnidipine, tracazolate hydrochloride and FPL 64176. IBMX (50 µM and/or 100 µM) was used as a positive control; DMSO (1%) was used as a negative control. Embryos were treated between 60–110 hpf prior to fixation and analysis for *vcanb* expression by whole-mount in situ hybridisation. Embryos were scored in accordance with two scoring systems, in order to assess the localisation (**A**) and the intensity (**B**) of *vcanb* ISH staining. (**A**) (i–iv) Scoring system used to assess the number of projections (p) with *vcanb* ISH staining. (v–viii) Charts showing the number of embryos that scored 0 p, 1 p, 2 p, or 3 p. (**B**) (i–iv) Charts showing the number of embryos that scored 0, 1, 2, or 3, according to the scoring system shown in **Figure 3A**. (**C**) (i–vi) Live DIC images of 110 hpf (or 90 hpf for FPL 64176-treated embryos) *adgrg6*tb233c mutants treated with the compounds shown above. Dorsal views with anterior to the left. (i'–vi') Lateral views of the inner ear of the embryos depicted in i–vi, showing rescue of pillar fusion (arrowheads) following treatment. (**D**) Measurements of the ear-to-ear width were taken from live embryos mounted dorsally and photographed at a focal plane that highlighted the largest visible dimensions (see **Figure 7—figure supplement 2**). Error bars represent the mean ± standard deviation. Combined data from two experimental repeats. Scale bars: 50 µm.

*Figure 7 continued on next page*

*Figure 7 continued*

DOI: https://doi.org/10.7554/eLife.44889.019

The following source data and figure supplements are available for figure 7:

**Source data 1.** Source data for the dose-response experiments shown in *Figure 7D*.
DOI: https://doi.org/10.7554/eLife.44889.023
**Figure supplement 1.** Additional dihydropyridines are able to downregulate otic *vcanb* expression in *adgrg6*[tb233c] mutant embryos.
DOI: https://doi.org/10.7554/eLife.44889.020
**Figure supplement 2.** Normalisation of ear width with respect to size of the head.
DOI: https://doi.org/10.7554/eLife.44889.021
**Figure supplement 2—source data 1.** Source data for the SSMD calculations shown in *Figure 7—figure supplement 2B*.
DOI: https://doi.org/10.7554/eLife.44889.024
**Figure supplement 3.** LD50 curves from the treatment of wild-type embryos from 60 to 110 hpf.
DOI: https://doi.org/10.7554/eLife.44889.022
**Figure supplement 3—source data 2.** Source data for the mortality counts shown in *Figure 7—figure supplement 3*.
DOI: https://doi.org/10.7554/eLife.44889.025

toxicity). Consistent with the *vcanb* scores for the number of projections stained, live DIC images of the inner ear revealed a dose-dependent rescue of projection fusion and pillar formation, which was greater at higher doses (*Figure 7C*). As *adgrg6*[tb233c] mutants have a swollen ear phenotype (*Geng et al., 2013*), measurements of the ear-to-ear width, normalised for size differences between individuals, were taken from photographs of live embryos mounted dorsally. The results showed a dose-dependent reduction in ear swelling with increased concentration of the four drugs (*Figure 7D*; *Figure 7—figure supplement 2*). LD50 concentrations were also determined for each of the four compounds and ranged from 19.2 µM (cilnidipine) to 51.7 µM (tracazolate hydrochloride) (*Figure 7—figure supplement 3*).

## Test for rescue of *vcanb* expression in the *fr24* allele: screen for Adgrg6-specific ligands

The initial screen was performed on the hypomorphic *tb233c* allele. We differentiated our hit compounds further by re-screening for *vcanb* expression in a strong *adgrg6* allele, *fr24* (*Figure 1B*), to identify compounds that could potentially interact directly with the Adgrg6 receptor itself. We predicted that any compounds able to rescue both alleles (such as IBMX at higher concentrations) are likely to act downstream of the receptor. On the other hand, hits that rescued *tb233c*, but were not able to rescue *fr24*, could potentially act as putative agonistic ligands for the Adgrg6 receptor. Of the 41 hit compounds able to rescue both *vcanb* and *mbp* in the *tb233c* allele, we identified 10 compounds that also rescued *vcanb* expression in the *fr24* screen (score sum 0–7 in *Table 1*, yellow), 12 compounds that gave a partial or inconclusive rescue (white), and 19 compounds that did not affect *vcanb* expression in the *fr24* screen (score sum nine in *Table 1*, grey). The first group (yellow) are presumed to act downstream of the Adgrg6 receptor, and include colforsin, which tested positive in all assays and is a known activator of adenylyl cyclase, supporting this interpretation (*Figure 8*). The last class (grey) are of particular interest as they represent candidates for molecules that may interact directly with the receptor. Examples of the difference in ability to rescue the two *adgrg6* alleles between the two classes can be seen in *Figure 8C*.

Interestingly, four of the 19 compounds in the last group are in the cluster of gedunin derivatives identified in *Figure 4* (cluster 2), with deoxygedunin being one of the top ten most potent drugs able to rescue the *tb233c* allele. The compound network shows that 38 compounds with structural similarity to the gedunins are represented in the two libraries (*Figures 5* and *8*). In the primary screens, 25/38 (66%) gedunin-related compounds affected *vcanb* expression to some extent (18 compounds in categories A–C and seven in D), nine compounds were inactive and four were toxic. The majority of the gedunin-related compounds that passed both rounds of retesting were later found also to rescue *mbp* expression (8/10, 80%). The shared structural characteristics of the gedunin group may give useful clues for candidate structures of agonistic ligands for Adgrg6. In summary, our study demonstrates a novel screening approach which, when combined with chemoinformatics

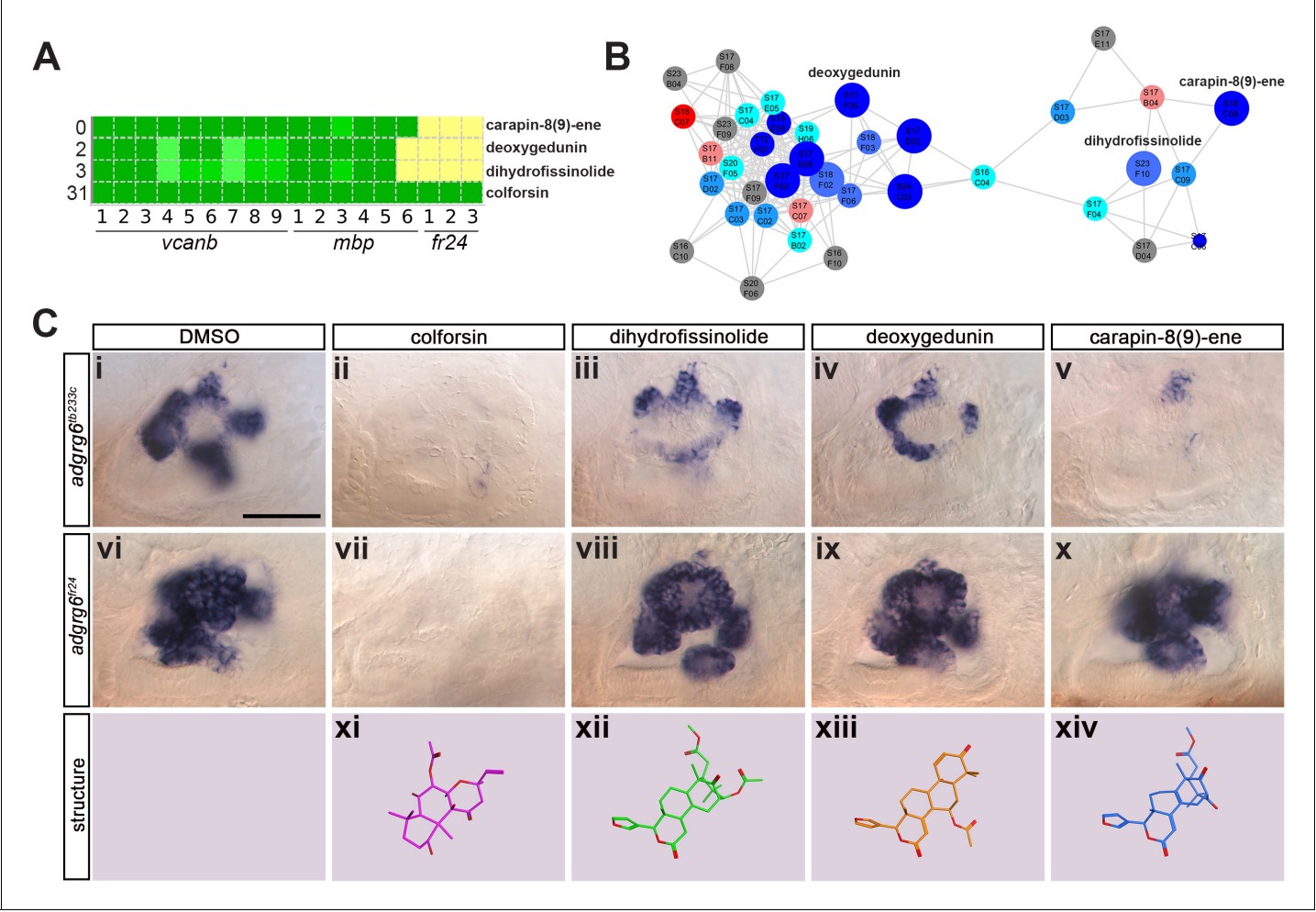

**Figure 8.** Assay for rescue of the *fr24* strong allele distinguishes compounds likely to rescue downstream, or at the level of, the Adgrg6 receptor. (A) Section of the heatmap in *Figure 5A* showing the results for colforsin, dihydrofissinolide, deoxygedunin and carapin-8(9)-ene. (B) Enlargement of the cluster containing gedunin-related compounds (cluster two in *Figures 4G* and *5B*), highlighting deoxygedunin, dihydrofissinolide and carapin-8(9)-ene. Compounds that rescued *mbp* expression are shown as larger nodes; compounds that did not rescue *mbp* expression are shown as smaller nodes. (C) (i–x) The inner ear at 4 dpf stained for *vcanb*. Lateral views; anterior to the left. Scale bar (applies to panels i–x): 50 μm. (i) *adgrg6^tb233c*/ DMSO mutant control. (ii – v) Treatment of *adgrg6^tb233c* mutants with the compounds at 25 μM indicated was able to rescue the *tb233c* mutant ear phenotype to variable degrees. (vi) *adgrg6^fr24*/DMSO mutant control. (vii–x) Treatment of *adgrg6^fr24* mutants with colforsin rescued otic *vcanb* expression in the *fr24* allele, whereas treatment with dihydrofissinolide, deoxygedunin and carapin-8(9)-ene was unable to rescue the *fr24* ear phenotype. (xi–xiv) Representation of the chemical structure of the four compounds tested. Note the structural similarity between deoxygedunin, dihydrofissinolide and carapin-8(9)-ene.

DOI: https://doi.org/10.7554/eLife.44889.026

analysis, is able to delineate both expected downstream rescuers of the Adgrg6 pathway and several candidates for drugs that may interact directly with the Adgrg6 receptor.

## Discussion

Adhesion GPCRs are critical regulators of development and disease, driving cell-cell and cell-ECM communications to elicit internal responses to extrinsic cues. This study set out to identify positive modulators of the Adgrg6 signalling pathway, a key regulator of myelination and inner ear development in the zebrafish embryo. Use of a whole-animal phenotypic (mutant rescue) screen gave the potential to identify compounds affecting the entire Adgrg6 pathway in the correct cellular context. We have used a simple in situ hybridisation approach to assay *vcanb* expression in the inner ear of

*adgrg6* mutants, exploiting an easily identifiable phenotype that could be scored manually. Following our primary screen of 3120 small molecules, we tested 89 hit compounds in a counter screen for rescue of the myelination defect in *adgrg6* mutant embryos. We identified 41 compounds that can both rescue *vcanb* expression in the inner ear and *mbp* expression in Schwann cells of *adgrg6* hypomorphic mutants, suggesting these are Adgrg6 pathway-specific modulators. Further analysis of a strong *adgrg6* allele, *fr24*, identified a subset of 19 compounds that are potential direct interactors of the Adgrg6 receptor. This analysis, combined with chemoinformatics analysis of the identified hit compounds, has identified clusters of compounds acting at different levels of the Adgrg6 pathway.

An optimal drug screening assay design identifies the maximum number of hit compounds with the minimum number of false positives and false negatives. Chemical screening assays using zebrafish range from simple morphology screens (*Yu et al., 2008*) through to high-tech, automated methods for quantitative image analysis (*Early et al., 2018*) or behavioural analysis (*Bruni et al., 2016*; *Rennekamp et al., 2016*) (reviewed in *Kalueff et al., 2016*). We used an in situ hybridisation screen to analyse gene expression changes, as this has the advantages of being scalable to different sized projects and relatively inexpensive to perform—with results that are stable and reproducible. Spatial resolution of staining patterns can be accurately scored: expression pattern screens have recently been used to identify small molecules that can induce subtle differences in gene expression domains along the pronephros (*Poureetezadi et al., 2016*) and in the somites (*Richter et al., 2017*). Although quantification of gene expression levels is less reliable with an enzymatic reaction compared with a fluorescent signal, we utilised the strong contrast between the high *vcanb* expression in the ear of *adgrg6* mutant fish compared with the low expression in a small dorsal region of the wild-type ear at four dpf to produce a robust scoring system for our phenotype rescue.

Relatively few zebrafish screens have been undertaken to identify compounds that can increase myelination (*Buckley et al., 2010*; *Early et al., 2018*) or restore myelination in neuropathy models (*Zada et al., 2016*), which is in part due to the complex distribution of glial cells in both the CNS and PNS. Performing the primary screen using an ear marker, *vcanb*, enabled us to bypass the difficulties of scoring and quantification of *mbp* staining on a large scale; instead, *mbp* expression was used as a counter screen on a limited number of cherry-picked hits. Contrary to the primary assays, which screened for down-regulation of *vcanb* expression, the counter screen assayed for up-regulation of *mbp* expression, enabling the identification of 21 false-positive compounds that down-regulate the expression of both genes (presumably by inhibiting transcription).

Determining the false-negative rate for any screen is difficult. In our assay we used only one concentration of compound (25 µM), so it is likely that some of the compounds that were toxic or showed no effect at 25 µM—and thus eliminated from our screen—would be effective at lower or higher concentrations, respectively. One possibility would be to run a parallel screen at a lower or higher concentration or use an alternative protocol with shorter incubation times, an approach that has recently proved successful at identifying different compounds influencing segmentation in zebrafish (*Richter et al., 2017*). Here, the compounds were found to be most active in the range of 10–50 µM, supporting our choice of 25 µM for the primary screen. However, increasing the number of replicates with different drug concentrations or assay conditions has significant implications on the cost and time taken to complete the screen, reducing the number of compounds analysed and the potential hits identified. An alternative method is to use a structural network to select and prioritise candidate compounds that have a similar structure to hit compounds and test these at different concentrations. Using this approach, we identified (±)-Bay K 8644 as an additional hit from the pyridine cluster when tested at a higher concentration. Our minimum estimate for the false-negative rate is 5%, based on the seven compounds (out of a total of 155) that were duplicated in both libraries and had a significantly different score after retesting, being classified as a hit in one library but not in the other. It is possible that this is due to differences in chemical purity from the different suppliers. Other false-negative compounds could include those that are unable to penetrate into the ear. Neomycin, for example, is toxic to the superficial hair cells of the lateral line system, but is ineffective on inner ear hair cells unless microinjected into the ear (*Buck et al., 2012*). Other compounds that we will have missed could include myelination-specific compounds, as the primary assay scored for the ear phenotype only. Given that several compounds were positive hits for the rescue of *vcanb* in the ear and negative for *mbp*, it is likely that tissue-specific functions of Adgrg6 are mediated through different downstream pathways or are stimulated by different ligands.

Our positive control compound, IBMX, was identified independently through our screen as a category A hit. The hit compound colforsin was found to be more potent and less toxic than the related control compound forskolin, and had the highest score in every assay, showing full rescue of the strong *fr24* allele. Both these observations highlight the robustness of the assay and the consistency of the scoring process. In total, the final number of hit compounds identified was similar in both the compound libraries screened, with 42 compounds identified from the Spectrum library (2.1%) and 27 compounds from the Tocris library (2.4%). These hit rates are comparable to those found in other similar screens (*Baxendale et al., 2012*; *Vettori et al., 2017*; *Wiley et al., 2017*).

Chemoinformatics analysis and visualisation of the results provided additional context to the identified hit compounds. The polar scatter plot displayed an initial overview of the results and allowed the identification of six different structurally-related clusters of active compounds with similar structure. The compound network focused the analysis on highly detailed similarity relationships inside each compound cluster, yielding a wealth of structure-activity relationship information that could prove very useful for any future optimisation of the identified hit compounds. Seven of the 41 hit compounds that rescued *vcanb* and *mbp* expression are $Ca^{2+}$-channel modulators. Six of these (nifedipine, cilnidipine, nitrendipine, nimodipine, efonidipine, niguldipine) belong to the chemical group of dihydropyridines (cluster 1), and initial investigations into compounds with similar structures identified four additional compounds in this class. Several dihydropyridines are known to have neuroprotective effects in mammalian models. Nimodipine, for example, has been shown to trigger remyelination in a mouse model of multiple sclerosis and to improve repair in peripheral nerve crush injuries in rats (*Schampel et al., 2017*; *Tang et al., 2015*), and some compounds have shown promise at clearing toxic proteins in animal models of neurodegeneration, including felodipine (*Siddiqi et al., 2019*) and nilvadipine (*Paris et al., 2011*). As dihydropyridines have been reported to inhibit cAMP phosphodiesterases (*Sharma et al., 1997*), protection of cAMP from degradation might be another mechanism whereby these molecules exert their ameliorating action on the *adgrg6* mutant phenotype.

Phenotypic screens are advantageous for assessing models of multifactorial pathological conditions, such as hereditary neuropathies and cancer (reviewed in *Baxendale et al., 2017*). However, one of the challenges for phenotypic screening is the identification of the specific target for any hit compound, as multiple pathways and different cell types can contribute to a positive read-out in the screening assay. Our aim was to identify compounds that could potentially interact directly with the Adgrg6 receptor. We were able to separate hit compounds into different groups based on their ability to rescue otic phenotypes caused by missense (*tb233c*) and nonsense (*fr24*) mutations. In total, we found 19 hits that could rescue *vcanb* expression and *mbp* expression in the *tb233c* allele, but were unable to rescue the *fr24* allele. We hypothesise that the *fr24* allele is unable to produce the full-length Adgrg6 protein including the CTF, and therefore any compounds that interact directly with the receptor would not be able to rescue any CTF-dependent function in this strong allele. Further analysis will be needed to determine whether any of these compounds can bind directly to the Adgrg6 receptor, for example by assessing stimulation of cAMP and direct binding in in vitro assays. However, this approach of using a combination of null and hypomorphic alleles in zebrafish whole-organism screening with the aim of identifying target-specific compounds is particularly exciting and one that the advent of CRISPR/Cas9 technology is placed to take full advantage of, since it is now possible to generate designer mutations in the zebrafish through homology-directed repair (*Hruscha et al., 2013*; *Hwang et al., 2013*; *Komor et al., 2016*).

It is of interest to note that one of the main groups of compounds identified as potential interactors of the receptor in the *fr24* screen is a cluster of gedunin derivatives (cluster 2). One of these compounds, deoxygedunin, has previously been identified as a TrkB agonist that has neuroprotective properties (*Nie et al., 2015*), can promote axon regeneration after nerve injury (*English et al., 2013*), and, interestingly, has been found to protect the vestibular ganglion from degeneration in mice mutant for *BDNF* (*Jang et al., 2010*). More recently, gedunin derivatives, including 3-α-DOG, have been shown to act as partial agonists for the closely related aGPCR, ADGRG1 (formerly GPR56) (*Stoveken et al., 2018*), a key regulator of myelination in both the CNS and PNS (*Ackerman et al., 2015*; *Ackerman et al., 2018*; *Giera et al., 2015*; *Salzman et al., 2016*). While further work will be necessary to determine if gedunin-type molecules can also bind and activate zebrafish Adgrg6 by interacting directly, these studies set a precedent for this type of interaction.

GPCRs can be modulated by the membrane lipid cholesterol, where interactions with the 7TM domain can provide plasticity for the receptors by altering their stability and structure (*Huang et al., 2018*; *Prasanna et al., 2016*). In addition, cholesterol can activate the hedgehog signalling pathway directly by binding to the extracellular domain of the GPCR Smoothened (*Huang et al., 2018*; *Luchetti et al., 2016*). Although cholesterol was not identified as a hit in our primary screen, we did identify two cholesterol-lowering drugs, ezetimibe (*Altmann et al., 2004*) and rosuvastatin (*Istvan and Deisenhofer, 2001*), as putative modulators of the Adgrg6 pathway. Whether these act by altering the activity of Adgrg6 through altering cholesterol levels remains to be determined.

In addition to the dihydropyridines (cluster 1) and the tetranortriterpenoid (gedunin-derived) compounds (cluster 2), there are also clusters of steroid hormones (danazol, hydroxyprogesterone, pregnenalone succinate, hydrocortisone hemisuccinate) and flavonoid compounds (baicalein, tangeritin, nobiletin, dimethylnobiletin, hexamethylquercetagetin). The flavonoids are a group of molecules with wide ranging activities, including anti-cancer (*Ma et al., 2015*) and neuroprotective properties (reviewed in *Braidy et al., 2017*). All four *O*-methylated flavonoids that rescued *vcanb* and *mbp* expression in *tb233c* mutants were also able to rescue *fr24* allele in our assay, suggesting that they act downstream of the Adgrg6 receptor.

Our screen identified 28 compounds that down-regulated *vcanb* expression, but did not rescue *mbp* expression, which may provide useful tools to manipulate semicircular canal formation in vivo. Versican and other chondroitin sulphate proteoglycans (CSPGs) are associated with a number of human pathologies; Versican overexpression has been shown to be strongly involved in inflammation, cancer progression and the development of lung disorders (reviewed in *Andersson-Sjöland et al., 2015*; *Ricciardelli et al., 2009*; *Wight et al., 2017*). CSPGs and hyaluronan are components of the inhibitory scar that forms at the site of injury after CNS damage, preventing axon regeneration (*Silver and Miller, 2004*). In addition, CSPGs have been shown to inhibit the ability of oligodendrocytes to remyelinate axons, a process that is reversed by reduction of CSPG levels (*Keough et al., 2016*; *Pendleton et al., 2013*). Whether the down-regulation of CSPGs to promote remyelination occurs via a similar mechanism to that involved in Adgrg6-regulated projection fusion remains to be determined. However, it is of interest that a key regulator of myelination, Adgrg1, has also been recently shown to reduce fibronectin deposition and inhibit cell-ECM signalling to prevent metastatic melanoma growth (*Millar et al., 2018*).

In conclusion, our data show that *vcanb* expression in the *adgrg6*^*tb233c* mutant ear provides a robust, easy-to-use screening tool to identify drugs that target the Adgrg6 pathway. In combination with the different alleles available for *adgrg6* in zebrafish, this in vivo platform provides an excellent opportunity to find hit compounds that may be specific for Adgrg6 in counter screens. These may provide a starting point for the development of therapeutic approaches towards human diseases where *ADGRG6* or myelination is affected. We have identified groups of structurally-related compounds that can rescue *adgrg6* mutant defects, including those that are likely to act downstream of the Adgrg6 pathway, and others that are candidates for interacting with the Adgrg6 receptor. The chemical analysis and structural comparison of the compounds shown to be putative Adgrg6 receptor agonists will contribute to the elucidation of the physical properties responsible for ligand binding and will provide further insight on the underlying mechanism of Adgrg6 signalling.

## Materials and methods

### Animals

Standard zebrafish husbandry methods were employed (*Westerfield, 2000*). To facilitate visualisation of in situ hybridisation (ISH) staining patterns, embryos of the *nacre* (*mitfa*^*w2/w2*) strain (ZDB-GENO-990423–18), which lack melanophores (*Lister et al., 1999*), but are phenotypically wild-type for expression of *vcanb* and *mbp*, were used as controls for all drug screening experiments. The wild-type strain used for dose-response experiments was London Wild Type (LWT). *adgrg6* mutant alleles used were *lau*^*tb233c* (formerly *bge*^*tb233c*) and *lau*^*fr24* (ZDB-GENE-070117–2161) (*Geng et al., 2013*; *Whitfield et al., 1996*), and were raised on a pigmented background. In all cases shown, mutant embryos are homozygous for the respective allele. The transgenic strain used for imaging in *Figure 1* and in the videos expresses GFP throughout the otic epithelium, and was a gift of Robert Knight (*Baxendale and Whitfield, 2016*). Prior to treatment, embryos were raised in E3 embryo

medium (*Westerfield, 2000*) at 28.5°C. We have used the term embryo throughout to refer to zebrafish embryos and larvae from 0 to 5 days post fertilisation (dpf).

## Compound storage, aliquoting and administration to embryos

Chemical compounds from the Tocriscreen Total library (Tocris, Batch #2884, 1120 compounds) and The Spectrum Collection (Microsource Discovery Systems, Batch #100122, 2000 compounds) were arrayed in MultiScreen-Mesh 96-well culture receiver trays (Millipore) in columns 2–11 and diluted to 25 µM in E3 medium for drug screening. Control wells contained either IBMX (3-isobutyl-1-methyl-xanthine, Sigma, 50 µM and 100 µM), DMSO (Sigma, 1% in E3) or E3, in columns 1 and 12 (see diagram of the plate layout in *Figure 2*). Wild-type (LWT and *nacre*) and homozygous *adgrg6*[tb233c] mutant embryos were raised to 50 hpf at 28.5°C in E3 medium, dechorionated manually with forceps, and then incubated at 20°C overnight to slow down development and facilitate timing of experimental treatments. This regime reduced ear swelling, but did not reduce otic *vcanb* levels, in mutant embryos. Embryos at the 60 hpf stage were aliquoted at three embryos per well into Multi-Screen-Mesh mesh-bottomed plates (Millipore) and transferred to the drug plate (receiver tray; see above). Assay plates were incubated at 28.5°C for 28 hr and the embryos were then transferred to 4% paraformaldehyde and stored at 4°C overnight. Embryos were bleached according to the standard protocol (*Thisse and Thisse, 2008*) and stored at −20°C in 100% methanol until required for ISH. Hits identified in the primary screen were rescreened using the same protocol. Selected compounds were purchased separately from Sigma (nifedipine, cilnidipine, nilvadipine), Sigma LOPAC Collection (nemadipine-A, felodipine, lercanidipine), Cayman Chemicals (FPL 64176) and Santa Cruz Biotechnology (tracazolate hydrochloride).

## Whole-mount in situ hybridisation analysis of gene expression

Digoxigenin-labelled RNA probes for *vcanb* (*Kang et al., 2004*) and *mbp* (*mbpa*) (*Brösamle and Halpern, 2002*) were prepared as recommended (Roche). Whole-mount ISH was performed using standard procedures (*Thisse and Thisse, 2008*), modified for the Biolane HTI 16V in situ robot (Intavis) and MultiScreen-Mesh mesh-bottomed plates to increase throughput (*Baxendale et al., 2012*). Stained embryos were scored manually by at least two people and any discrepancies between the results were re-analysed. For the dose-response data the results were blinded and re-scored to check for consistency.

## Scoring systems for *vcanb* and *mbp* expression

To score the efficacy of the drugs in down-regulating *vcanb* mRNA levels, a scoring system from 0 to 3 was used, with 0 being the score for a very efficient drug (a 'hit') that can suppress *vcanb* expression back to almost wild-type levels, and 3 the score for a drug that did not have any effect on *vcanb* mRNA levels expressed in the *adgrg6* mutant ear. Scores 1 and 2 were given to drugs that showed an ability to down-regulate *vcanb* expression to some extent, with 1 given for a stronger down-regulation than 2 (*Figure 3A*). Drugs were then classified into categories A–E, according to the combined score from the three embryos treated with each drug (*Figure 3B*).

For the *mbp* counter screen (*Figure 4A,B*), a score of 3 was used for embryos where *mbp* mRNA expression was rescued to wild-type levels, a score of 2 for embryos that showed some *mbp* expression around the PLLg (weaker than wild-type levels) and a score of 1 in cases where the *mbp* expression was identical to the one seen in untreated *adgrg6*[tb233c] mutants (i.e. lacking *mbp* expression around the PLLg). The fact that *mbp* expression is not missing altogether from other areas of the PNS in *adgrg6*[tb233c] mutants allowed us to use a score of 0 in cases where *mbp* expression levels were lower than those typically seen in *adgrg6*[tb233c] mutants.

## Hit selection

Drugs categorised as A or B were considered successful and were cherry-picked into new drug assay plates for further testing. Drugs categorised as C were potentially interesting and a few were used to complete a 96-well cherry-pick plate (37/96 compounds). Drugs categorised as D and E were considered to show incomplete or no inhibition of *vcanb* expression, respectively. Drugs from category F caused severe developmental abnormalities, heart oedema, brain oedema or death at the end of the treatment and therefore were characterised as toxic. Category G represented drugs that were

potentially corrosive, as no fish were found in these wells at the end of the treatment, although this could also have resulted from death of the embryos followed by digestion by microorganisms, or through experimental error. Drugs that fell into any of the categories D–G and most from category C (59) were eliminated from the assay and were not followed further. In addition, 10 compounds from category A and B were unable to be tested further due to compound availability issues.

Compounds taken forward for secondary assays were chosen by two criteria: 1. A final average score for three replicates (total nine embryos) with a category A–C; 2. No individual score >7. In total, 91 compounds were picked for the counter screen, including two that were present in both libraries, resulting in 89 individual compounds. The screening pipeline is shown in *Figure 4G* and the subsequent grouping of compounds is described in *Figures 4*, *5*, *6* and *8*.

## Dose-response and LD50 assays

Selected compounds were tested in dose-response assays. In order to assess the ear swelling in drug-treated *adgrg6*$^{tb233c}$ mutant embryos, the ear-to-ear width was measured from photographs of live embryos mounted dorsally, and normalised for the size of the head, using CELLB software (for details, see *Figure 7—figure supplement 2*).

An LD50 curve was plotted for the adjusted exposure time (60–110 hpf), using 16 LWT wild-type embryos (biological replicates) per concentration. To avoid cross-contamination from dead embryos, each wild-type (LWT) embryo was kept in a separate well of a 96-well plate. At the end of each treatment, the number of dead embryos (no heartbeat for 10 s) was recorded.

## Microscopy and photography

Still images of live embryos were taken using an Olympus BX51 microscope, C3030ZOOM camera and CELLB software, and assembled with Adobe Photoshop. All micrographs are lateral views with anterior towards the left and dorsal towards the top, unless otherwise stated. For archiving, fixed and stained embryos were imaged in MultiScreen-Mesh plates containing 50% glycerol, using a Nikon AZ100 microscope with an automated stage (Prior Scientific). A compressed in-focus image was generated using the NIS-Elements Extended Depth of Focus software (Nikon).

Time-lapse imaging of live embryos was performed on a Zeiss Z.1 light-sheet microscope. *adgrg6*$^{fr24}$ homozygous mutant embryos in a transgenic background (see Animals) were mounted at 60 hpf in 0.7% agarose with anaesthetic (MS-222; 160 µg/ml) and 0.003% PTU (to prevent pigment formation). Images were taken of a dorsal view of the ear every 5 min (200 z-slices, 1 µm sections). A control time-lapse of a wild-type sibling embryo (images taken at 10 min intervals) was taken on a separate day. Images were cropped and a subset of z-slices through the anterior (*adgrg6*$^{fr24}$) and posterior (phenotypically wild-type sibling) projections were used to make Maximum Intensity Projection videos of projection fusion in the wild-type sibling and the swollen projections in *adgrg6*$^{fr24}$ mutant embryo. The two videos do not correspond exactly to the same developmental stage.

## Chemoinformatics analysis and data visualisation

Chemical structures of the library compounds represented as SMILES (*Weininger, 1988*) were obtained from vendor catalogues. Molecules were standardised using the wash procedure of MOE (Chemical Computing Group Inc, Molecular Operating Environment (MOE), Montréal, QC, 2011), accessed through KNIME (*Berthold et al., 2009*). Standardised molecules were analysed using RDKit (RDKit: Open-Source Cheminformatics, http://www.rdkit.org/, accessed 06 Nov. 2018) in Python (Python Software Foundation: Python language reference, version 3, https://www.python.org/, accessed 06 Nov. 2018). Morgan fingerprints of radius 2 (equivalent to ECFP4 [*Rogers and Hahn, 2010*]) were computed for each compound. Compound similarity was calculated using the Tanimoto coefficient (*Willett et al., 1998*) of the fingerprints using the scikit-learn library (*Pedregosa et al., 2011*). Based on the similarity matrix between all compound pairs, a dendrogram was obtained using the SciPy library (SciPy: Open Source Scientific Tools for Python, http://www.scipy.org/, accessed 06 Nov. 2018). The polar scatterplot was created using the matplotlib library (printed version) (*Hunter, 2007*) and plotly (interactive version) (Plotly Technologies Inc, Collaborative data science, Plotly, Montréal, QC, 2015). To identify duplicated molecules, the InChIKey (*Heller et al., 2015*) was computed for each compound and all pairs of compounds were checked for identical InChIKeys. To create the compound network, the similarity matrix computed for the dendrogram

was transformed into an adjacency matrix using a threshold value of 0.5; that is, compounds with a similarity value over 0.5 are connected with an edge. The network visualisation was created using Cytoscape (*Shannon et al., 2003*).

## Statistical analysis

Statistical analyses were performed using GraphPad Prism version 7 for Mac, GraphPad Software, La Jolla California USA, www.graphpad.com. The Strictly Standardised Mean Difference (SSMD, $\beta$) (*Zhang, 2007*) for the dose-response measurements in *Figure 7D* was calculated using the formula:

$$\beta = \frac{\mu_1 - \mu_2}{\sqrt{\sigma_1^2 + \sigma_2^2}}$$

# Acknowledgements

We thank a number of undergraduate and MSc project students who contributed to early stages of the primary screens described here, especially D Butler, who helped with establishing the *mbp* secondary screening protocol. F-S Geng tested the initial screening protocol on wild-type embryos. We thank J-P Ashton, S Burbridge, M Marzo and N van Hateren for technical support, D Lambert for discussion and the Sheffield aquarium staff for expert care of the zebrafish.

# Additional information

## Competing interests

Tanya T Whitfield: Reviewing editor, *eLife*. The other authors declare that no competing interests exist.

## Funding

| Funder | Grant reference number | Author |
| --- | --- | --- |
| Biotechnology and Biological Sciences Research Council | Project grant BB/J003050/1 | Sarah Baxendale<br>Tanya T Whitfield |
| University of Sheffield | PhD studentship 314420 | Elvira Diamantopoulou<br>Tanya T Whitfield |
| Medical Research Council | G0802527 | Sarah Baxendale<br>Celia J Holdsworth<br>Leila Abbas<br>Tanya T Whitfield |
| European Union Seventh Framework Programme | Grant agreement no. 612347 | Antonio de la Vega de León<br>Valerie J Gillet |
| Biotechnology and Biological Sciences Research Council | BB/R50581X/1 | Sarah Baxendale<br>Anzar Asad<br>Giselle R Wiggin<br>Tanya T Whitfield |
| Wellcome | VIP award 084551 | Leila Abbas<br>Tanya T Whitfield |
| Medical Research Council | G0700091 | Sarah Baxendale<br>Celia J Holdsworth<br>Leila Abbas<br>Tanya T Whitfield |
| Biotechnology and Biological Sciences Research Council | Project grant BB/M01021X/1 | Sarah Baxendale<br>Tanya T Whitfield |
| Biotechnology and Biological Sciences Research Council | ALERT14 equipment award BB/M012522/1 | Sarah Baxendale<br>Tanya T Whitfield |

The funders had no role in study design, data collection and interpretation, or the decision to submit the work for publication.

## Author contributions
Elvira Diamantopoulou, Data curation, Formal analysis, Validation, Investigation, Visualization, Writing—original draft, Writing—review and editing; Sarah Baxendale, Conceptualization, Data curation, Formal analysis, Supervision, Funding acquisition, Validation, Investigation, Visualization, Writing—original draft, Project administration, Writing—review and editing; Antonio de la Vega de León, Data curation, Formal analysis, Validation, Visualization, Writing—review and editing, Chemoinformatics analysis; Anzar Asad, Data curation, Validation, Investigation, Visualization, Writing—review and editing; Celia J Holdsworth, Investigation; Leila Abbas, Conceptualization, Investigation; Valerie J Gillet, Visualization, Writing—review and editing, Chemoinformatics analysis; Giselle R Wiggin, Conceptualization, Supervision, Writing—review and editing; Tanya T Whitfield, Conceptualization, Formal analysis, Supervision, Funding acquisition, Writing—original draft, Project administration, Writing—review and editing

## Author ORCIDs
Elvira Diamantopoulou (iD) http://orcid.org/0000-0002-9336-7965
Sarah Baxendale (iD) http://orcid.org/0000-0002-6760-9457
Antonio de la Vega de León (iD) https://orcid.org/0000-0003-0927-2099
Valerie J Gillet (iD) https://orcid.org/0000-0002-8403-3111
Giselle R Wiggin (iD) https://orcid.org/0000-0003-4436-9208
Tanya T Whitfield (iD) https://orcid.org/0000-0003-1575-1504

## Ethics
Animal experimentation: All animal work was performed under licence from the UK Home Office (P66302E4E), and approved by the University of Sheffield Ethical Review Committee (ASPA Ethical Review Process).

## Decision letter and Author response
Decision letter https://doi.org/10.7554/eLife.44889.029
Author response https://doi.org/10.7554/eLife.44889.030

# Additional files

## Supplementary files
• Supplementary file 1. List of the 89 hit compounds that rescued the expression of *vcanb* in *adgrg6*$^{tb233c}$ mutants and were followed up by *mbp* counter screens. The table includes the plate and well position of each compound, along with known activities and the raw data scores from nine *adgrg6*$^{tb233c}$ embryos in the *vcanb* assay (v1–v9), from six *adgrg6*$^{tb233c}$ embryos in the *mbp* assay (m1–m6) and from three *adgrg6*$^{fr24}$ embryos in the *fr24* (fr1–3) assay. Abbreviations: DE, dead embryo; ND, no data; S, Spectrum; T, Tocris. *Deoxygedunin: (*Jang et al., 2010*); Nobiletin: (*Cheng et al., 2016*); Angolensin (R): (*Weisman et al., 2006*); Sinensetin: (*Kang et al., 2015*); Larixol acetate: (*Urban et al., 2016*); Gedunin: (*Hieronymus et al., 2006*; *Subramani et al., 2017*).
DOI: https://doi.org/10.7554/eLife.44889.027
• Transparent reporting form
DOI: https://doi.org/10.7554/eLife.44889.028

## Data availability
All data generated or analysed during this study are included in the manuscript and supporting files. Source data files have been provided for Table 1 and Figure 1-figure supplement 1, Figure 3, Figure 7 and Figure 7-figure supplements. Links to interactive files are given in the manuscript and in a supplementary file.

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
