## [Decision Letter]

Thank you for submitting your article "Identification of compounds that rescue otic and myelination defects in the zebrafish *adgrg6 (gpr126*) mutant" for consideration by *eLife*. Your article has been reviewed by three peer reviewers, one of whom is a member of our Board of Reviewing Editors, and the evaluation has been overseen by Didier Stainier as the Senior Editor. The following individual involved in review of your submission has agreed to reveal their identity: David W Raible (Reviewer #3).

The reviewers have discussed the reviews with one another and the Reviewing Editor has drafted this decision to help you prepare a revised submission.

Summary:

In "Identification of compounds that rescue otic and myelination defects in the zebrafish *adgrg6 (gpr126*) mutant," Diamantopoulou, Baxendale et al., present the results of a set of screening and validation protocols that identify classes of compounds that rescue phenotypes associated with disruption to the adhesion gpcr *adgrg6*, aka *gpr126*. Disruption to *gpr126* in zebrafish leads to defects in inner ear morphogenesis and peripheral nerve myelination by Schwann cells. The availability of *gpr126* mutant alleles (missense and nonsense mutations) has allowed construction of a clever primary screen to identify regulators of mutant-related *vcanb* expression in the ear), and a secondary screen to identify compounds that do/don't also affect myelination. Extensive follow-up validation and categorisation has been carried out, culminating in the identification of compounds that may represent direct agonists of the aGPCR itself, which would be of great interest. In general the data are of a very high standard, the manuscript is well written, and all major claims are well substantiated by observations.

The reviewers and handling editor agreed that the major strength of this current work is as an excellent exemplar of how to execute and follow-up chemical genetic screens in zebrafish, rather than in the provision of specific insights into mechanisms of ear/glial cell biology. As such, we would encourage a revision in the Tools and Resources format. All reviewers and the handling editor agree that essential revisions outlined below should strengthen the manuscript and be achievable within the standard revision period.

Essential revisions:

Additional analyses are required related to the screen and follow-up analyses

1) The network analysis is a nice way to categorize and display the different compounds and reveal possible commonalities. A good test of whether these commonalities are related to function would be to test additional related compounds not in the original panels. This is the principal request for new experimental data.

2) The identification of 90 compounds out of 3000 (3%) seems rather high, suggesting that either the assays have a potentially high rate of false positive hits or there is the potential for nonspecific action revealed by the assays. Could the authors please comment on this?

Similarly, it would be nice if the authors provided a summary of the reproducibility of the *mbp* staining assay. These data are in Figure 5A, but the results appear more variable than the *vcanb* assay. Are there possible false-positives here?

3) The dose-response functions provide a good measure of concentration dependent activity. It would have been nice if the controls were repeated with each dose-response function (they appear to be the same data across all graphs in Figure 7C), as these would give a better representation of the reliability of the assay. It would be good if the z-prime was calculated for this assay.

4) The authors describe the similarities in results between 155 compounds found between the two libraries. It would be useful to report the statistical correlation between the two sets of scores as a measure of reliability.

A little more clarity in presentation/ writing is required in a few places throughout the manuscript.

5) Following how compounds advanced through the testing funnel is a little hard to follow. It would be good if there were a way to easily represent this such as a flow chart or other graphic. For example it is not clear how the 91 compounds were identified by the secondary screen (Figure 4). Why were only some compounds advanced for retesting (227/297 compounds)? What criteria advanced them into secondary screening? I can't see an easy way to get 91 compounds from categories in Figure 4A (I get 88 compounds from retest categories A and B).

6) The flow of paragraphs (subsection “Compounds that can rescue both inner ear and myelination defects”) is not quite right in that the first paragraph talks about selecting two groups for further analysis and then the next paragraph also mentions compounds chosen for further study. This second announcement sounds independent of the first one. Perhaps in the second paragraph the authors could state, "An additional selection…" or a 'third group'.

7) The authors mention that the *fr24* allele encodes a highly truncated protein, but I am not sure I see any evidence of this, or in the previous cited paper. One might expect, instead, that this allele would lead to nonsense mediated mRNA decay, but even if this did not occur, it is rather hard to see how the predicted protein/ peptide fragment could be presented in a physiological manner. The authors should either remove, qualify, or amend this statement.

8) It is not clear why the one screening concentration was selected? Can the authors explain, and discuss further in the context of screen design and execution.

9) In Figure 8B the use of gray lines makes it difficult to visualize the gray circles.

10) What is the significance of the yellow colors in the table shown in Figure 4A?

11) For dose-response curves, were measurements done blinded?

12) It is not clear what the two sets of graphs are in Figure 7A – are these replicates?

---

## [Author Response]

Essential revisions:Additional analyses are required related to the screen and follow-up analyses1) The network analysis is a nice way to categorize and display the different compounds and reveal possible commonalities. A good test of whether these commonalities are related to function would be to test additional related compounds not in the original panels. This is the principal request for new experimental data.

We thank the reviewers for their suggestion and agree the network analysis is useful for the display of structural commonalities and as a basis for determining structure-function relationships. To test additional related compounds, we have taken the dihydropyridine network analysed in Figure 6 as a starting point and have selected four additional dihydropyridines that are structurally related to nifedipine, but which were not represented in the Spectrum or Tocris libraries. The new compounds tested are nilvadipine, nemadipine A, felodipine and lercanidipine, which are all drugs of the type used to treat hypertension and are available from Sigma. We have tested these additional compounds in *adgrg6* rescue assays; encouragingly, all four can also rescue the down-regulation of otic *vcanb* expression in *adgrg6^tb233c^* mutant embryos.

Nilvaldipine has a high structural similarity to nifedipine (Tanimoto similarity coefficient >0.5), and we have modified the network in Figure 6B to show this (green circle). Nilvadipine was initially tested at 22.5 µM and showed a partial rescue of otic *vcanb* expression and a strong rescue of *mbp* expression around the posterior lateral line ganglion (new panels, Figure 6C vii, xiv). We also demonstrated a dose-dependent rescue of otic *vcanb* expression with higher concentrations of nilvadipine, with a reduction of expression to wild-type levels after treatment with 50.6 µM (new Figure 7—figure supplement 1).

The additional three new compounds showed a range of effects at 25 µM. Nemadipine A fully rescued otic *vcanb* expression, felodipine partially rescued and lercanidipine failed to rescue *vcanb* expression (Figure 7—figure supplement 1). We next tested if a higher drug concentration was effective for the latter two compounds. At 40 µM, felodipine still showed partial rescue of *vcanb* , but lercanidipine was able to rescue *vcanb* expression at 50 µM (Figure 7—figure supplement 1).

We also tested the Tocris compound (±)-Bay K 8644, which is included in the same network as nifedipine but did not score as a hit in the primary screen (25 µM). At a higher concentration (40 µM), we found that (±)-Bay K 8644 was now able to rescue otic *vcanb* expression (Figure 7—figure supplement 1). The testing of (±)-Bay K 8644 highlights how the network analysis can aid in prioritising compounds to test at alternative doses. These data are also relevant for our response to point 8 (see below).

We have described these new data in the Results (subsection “Compounds that can rescue both inner ear and myelination defects”, last two paragraphs) and have also modified the Discussion (fourth and sixth paragraphs). See also modified Figure 6, and new Figure 7—figure supplement 1. Information on the source for the additional compounds is included in the Materials and methods (subsection “Compound storage, aliquoting and administration to embryos”).

2) The identification of 90 compounds out of 3000 (3%) seems rather high, suggesting that either the assays have a potentially high rate of false positive hits or there is the potential for nonspecific action revealed by the assays. Could the authors please comment on this?

This point also relates to point 5 asking for further explanation of the screening funnel for how compounds were selected. To address this, we have included a new graphical summary in Figure 4G, which we hope will clarify the screening results and outline our strategy to remove false positives and those compounds with non-specific action.

To summarise, the list of 91 hit compounds included two compounds that were present in both the Spectrum and the Tocris libraries, and therefore the total number of unique compounds was 89 out of a total of 3120 compounds (or 2965 excluding all duplicates). The counter screen for up-regulation of *mbp* expression also identified compounds that reduced transcription of both *vcanb* and *mbp*—potential false positives—and this reduced the number of verified hits with *vcanb* to 68 (2.3% of 2965). This counter screen also used two independent phenotypes (ear and myelination) to identify any hits that were not specific for the Adgrg6 pathway, and this reduced the number of hit compounds that could rescue both *vcanb* and *mbp* expression to 41 (1.4%). We further tested whether compounds were unable to rescue the strong allele as potential interactors with the Adgrg6 protein and identified 19 compounds (0.6%) in this final assay.

We therefore feel that the overall hit rate for Adgrg6 modulators of (1.4%) is not higher than we would expect in comparison to published phenotypic screens. Wiley, Redfield and Zon (2017) reviewed the percentage of hits identified from a variety of drug screens in zebrafish and our hit rate is well within the range seen in other phenotypic screens (Discussion, fifth paragraph). We have now cited this study in the Discussion (fifth paragraph). It should also be taken into account that the Tocris and Spectrum libraries include known bioactives and FDA-approved drugs, which are more likely to have activity in an in vivo screen than a completely novel synthetic compound library, where the hit rate would be expected to be lower. We expect the list of hits will include compounds, like colforsin and our control compound IBMX, acting downstream of Adgrg6 signalling by increasing cAMP levels, and we would expect there to be a number of compounds with similar activity in these two libraries.

We therefore prefer to retain the higher rate of false positives after the primary screen and then use the retests and counter screens to validate hits. The screening design could be made more robust by increasing the number of embryos screened or performing multiple repeats; however, in our experience, the time and financial cost of screening in duplicate or triplicate at the primary screen stage in order to identify fewer primary hits is not cost effective and would reduce the number of compounds screened overall. This is already discussed in the text (Discussion, fourth paragraph).

Similarly, it would be nice if the authors provided a summary of the reproducibility of the mbp staining assay. These data are in Figure 5A, but the results appear more variable than the vcanb assay. Are there possible false-positives here?

The difference in *mbp* staining between wild-type and mutant embryos is highly reproducible, even when using the hypomorphic allele. Although expression in the PNS as a whole in the *adgrg6^tb233c^* mutant is variable, expression around the posterior lateral line ganglion is one of the few regions that is consistently absent, as discussed in the Results (subsection “Choice of markers for an in situ hybridisation-based screen: otic *vcanb* expression as a robust readout”, last paragraph). To quantify this, and to illustrate the variability of *mbp* staining intensity around the posterior ganglion between individual wild-type embryos and *adgrg6* mutants, we have mounted individual embryos for imaging. In dorsally mounted embryos, it is possible to distinguish the peripheral *mbp* staining (decreased in mutants) from the CNS staining (unaffected in mutants). We have taken 10 *adgrg6^tb233c^* mutant embryos and 10 siblings from the same in situ hybridisation experiment and analysed the left and right posterior lateral line ganglion region from each embryo. After thresholding for colour using the HSB settings in Fiji we have plotted the percentage area of *mbp* staining. The results show a statistically significant reduction in staining area in mutant embryos. We have included these data in a new supplementary figure (Figure 1—figure supplement 1).

In the screen, some of the *mbp* staining results show variability between individual embryos treated with a given concentration of drug (Figure 5A), but as we are looking for up-regulation of gene expression in this assay, we would expect to have more false-negatives than false-positives at this stage. Note also that the individual mounting required for the quantitative analysis described above is not practicable for a large-scale primary screen.

3) The dose-response functions provide a good measure of concentration dependent activity. It would have been nice if the controls were repeated with each dose-response function (they appear to be the same data across all graphs in Figure 7C), as these would give a better representation of the reliability of the assay.

To clarify, two different dose response assays were performed in separate 96-well plates and each plate included a set of controls. One plate screened increasing doses of FPL 64176 using an optimised assay window of 60-90 hpf and the second plate had increasing doses of nifedipine, tracazolate and cilnidipine using the time window of 60-110 hpf. There are therefore two sets of controls, one for the 60-90 hpf assay and another set for the 60-110 hpf assay. The experiment was repeated for both assay plates, and so the controls contain data points from two replicates. To make this information clearer we have rearranged this figure and legend so that the FPL 64176 data including controls is on one graph (Figure 7Di) and the other three compounds, with their controls, are on a second graph (Figure 7Dii).

It would be good if the z-prime was calculated for this assay.

In general, z-prime scores are not suitable for in vivo phenotypic screens, such as this one, where the results are initially qualitative and then assigned to an arbitrary numerical scoring system, or where the sample size (*n*) is low. In order to have a significant z-prime score it is necessary to have well-separated distributions (large difference between the means, small standard deviations) between the positive and negative control.

Thus, instead of a z-prime score, we have used our quantitative data from the dose response assay to determine the SSMD (Strictly Standardised Mean Difference, β) for assessing assay quality. This method has been used in RNAi screens (Zhang, 2007). We used the data from Figure 7D (ear width measurements) and the SSMD formula for two independent groups,

β=μ1-μ2σ12+σ22

where µ_1_ is the mean ear width distance of the *adgrg6^tb233c^* mutants treated with DMSO (negative control) and µ_2_ is the mean ear width distance of the *adgrg6^tb233c^* mutants treated with IBMX (positive control), σ12is the variance of the negative control and σ22 is the variance of the positive control. This generated an SSMD score of 2.811 for the controls on the plate with nifedipine, cilnidipine and tracazolate, which demonstrates that this assay has a strong threshold for hit detection. Scores for the hit compounds, compared to the negative control, were also strong at the higher concentrations (nifedipine score 2.48 at 33.7 μM, cilnidipine score 4.19 at 22.5 μM and tracazolate score 3.5 at 22.5 μM). The controls for the FPL 64176 plate have a slightly lower score of 2.03 after a 24-hour treatment, compared with the 48-hour treatment on the other plate. FPL 64176 had a similar score of 2.04 at 6.7 μM. We have included all the scores in a modified Figure 7—figure supplement 2 and included the method in the statistics section of the Materials and methods (subsection “Statistical analysis”) (see also revised Figure 7—figure supplement 2—source data 1).

4) The authors describe the similarities in results between 155 compounds found between the two libraries. It would be useful to report the statistical correlation between the two sets of scores as a measure of reliability.

As mentioned in response to point 3, our primary assay scores are not suited to statistical correlation as they are not quantitative. We had analysed the duplicated compounds in detail and had already discussed the few differences we found in the text (subsection “Validation of the primary screen: retesting, comparison with control compounds and analysis of duplicates”, third paragraph).

A little more clarity in presentation/ writing is required in a few places throughout the manuscript.5) Following how compounds advanced through the testing funnel is a little hard to follow. It would be good if there were a way to easily represent this such as a flow chart or other graphic. For example it is not clear how the 91 compounds were identified by the secondary screen (Figure 4). Why were only some compounds advanced for retesting (227/297 compounds)? What criteria advanced them into secondary screening? I can't see an easy way to get 91 compounds from categories in Figure 4A (I get 88 compounds from retest categories A and B).

As discussed in response to point 2, we have now added further clarification of our screening funnel in the Materials and methods section and also made a new panel summarising the hit selection steps for Figure 4G. Briefly, from the primary screen, the list of hits included those in categories A–C. While we were interested in those that had the strongest rescue (A, B categories) we also wanted to determine how many compounds in category C would come through the retests as a hit. We therefore made up the cherry-pick ‘hit’ plates with extra compounds from the C category, although not all of category C compounds were retested. In addition, 10 compounds from the B category were not followed up due to a lack of compound availability in the cherry-pick plates—this is mentioned in the Materials and methods (subsection “Hit selection”). The table in the original Figure 4A confirmed that the majority of compounds from category C retested as category D (non-hit). However, we have taken the reviewers’ comments on board and have removed this table and replaced it with a schematic of the hit selection process, which we hope will be clearer for the reader. We have moved some of the details of the screening process from the Results section to the Materials and methods section. The revised text can be found in the Results (subsection “Validation of the primary screen: retesting, comparison with control compounds and 286 analysis of duplicates”, first paragraph) and Materials and methods (subsection “Hit selection”). See also the new Figure 4G.

6) The flow of paragraphs (subsection “Compounds that can rescue both inner ear and myelination defects”) is not quite right in that the first paragraph talks about selecting two groups for further analysis and then the next paragraph also mentions compounds chosen for further study. This second announcement sounds independent of the first one. Perhaps in the second paragraph the authors could state, "An additional selection…" or a 'third group'.

We have now amended this section and moved the discussion of the heatmap to the previous section (subsection “Secondary screen for rescue of *mbp* expression, and identification of false positives”, last paragraph). We have also clarified which of the selected compounds were from the two groups that correspond to the clusters in Figure 4D and Figure 5B and which were chosen separately and hopefully this is now clearer (subsection “Compounds that can rescue both inner ear and myelination defects”).

7) The authors mention that the fr24 allele encodes a highly truncated protein, but I am not sure I see any evidence of this, or in the previous cited paper. One might expect, instead, that this allele would lead to nonsense mediated mRNA decay, but even if this did not occur, it is rather hard to see how the predicted protein/ peptide fragment could be presented in a physiological manner. The authors should either remove, qualify, or amend this statement.

The reviewers are correct that we do not have any experimental evidence for the presence or absence of a truncated protein product in the *fr24* allele. We have been careful to explain this in the text (subsection “Choice of markers for an in situ hybridisation-based screen: otic *vcanb* expression as a robust readout”, second paragraph) and have checked all other references in the manuscript to *fr24* to remove any ambiguity. As shown in Figure 1 and in our previous study (Geng et al., 2013), the *fr24* allele predicts a stop codon L463X which is within the extracellular domain of the protein, N-terminal to the hormone-binding domain and GAIN domain.

Although nonsense-mediated mRNA decay is a possibility, we have previously tested the presence of the *adgrg6* transcript in *adgrg6^fr24^* mutants by in situ hybridisation. In some regions expression is weaker, but expression in the swollen projections in the ear is strong (Geng et al., 2013). Given that some transcript is produced, it is possible that it is translated into protein. If the truncated N-terminal fragment (amino acids 1–462) is expressed as protein it would contain the signal peptide needed to export the peptide and also the CUB and PTX domains. We note that another truncating *adgrg6* allele, *st49*, is known to produce a slightly larger peptide, which retains the hormone binding domain and is biologically active in vivo, being able to restore radial sorting of axons in *adgrg6* mutants, so there is precedent for the production of truncated Adgrg6 peptides (Petersen et al., 2015). This is now discussed in the aforementioned paragraph.

However, the purpose of using the *fr24* allele in the current study is to distinguish between compounds that can activate downstream signalling of the Adgrg6 pathway (which are active whether the Adgrg6 protein is present or not) and those that potentially interact with the Adgrg6 protein (active in the weaker *adgrg6^tb233c^* allele; inactive in *fr24*). In the latter case, even if an N-terminal peptide is produced and is active in *fr24* mutants, the mutation predicts that the transmembrane domain necessary for G protein coupling to activate downstream signalling will be missing.

8) It is not clear why the one screening concentration was selected? Can the authors explain, and discuss further in the context of screen design and execution.

The choice of compound concentration for screening is always a compromise between efficacy and toxicity. The concentration 25 μM was chosen empirically after analysing results from previous screens performed at both 10 μM (Baxendale et al., 2012) and 25 μM (pilot data from another screen, unpublished). These data suggested that the number of toxic compounds did not increase significantly at the higher concentration. 25 μM is within the range frequently used in zebrafish assays (Richter et al., 2017; Wiley et al., 2017). This is discussed in the fourth paragraph of the Discussion.

As highlighted in our response to point 1, the new Figure 7—figure supplement 1 now includes a panel of dihydropyridine compounds that were only partially active in the assay at 25 μM, but were active at higher concentrations of 40 or 50 μM. Lowering the concentration in the primary screen to 10 μM would have resulted in more hit compounds being missed, including nifedipine, based on our dose-response data in Figure 7B. Screening the compounds in duplicate with two different concentrations would have reduced the overall number of compounds we could screen, taking in to account the number of embryos available and the cost of other reagents. One potential compromise would be to take all the compounds that proved to be toxic at 25 μM and re-screen these cherry-picked compounds at a lower concentration.

9) In Figure 8B the use of gray lines makes it difficult to visualize the gray circles.

The colour of the lines has been modified to make the network easier to visualise in Figures 6, 8 and Figure 7—figure supplement 1. The entire network is also available as an interactive version, where individual networks can be enlarged to aid visualisation.

10) What is the significance of the yellow colors in the table shown in Figure 4A?

The yellow colours in the previous Figure 4A highlighted the categories with the highest numbers of hit compounds after retesting. However, after consideration of the reviewers’ points 2 and 5 we have decided to remove this table and have replaced it with a schematic of the hit selection process that also still includes the numbers of compounds in each category at each stage (Figure 4G). See also our response to point 5 above.

11) For dose-response curves, were measurements done blinded?

The ear measurements from the ear swelling dose response assays were not done blinded. In general, all plates were scored twice and by two different scorers. For the dose-response assay the reviewer refers to, the in situ hybridisation results for the *vcanb* staining were still available; we went back to the original plates and re-ordered them to do a retrospective blinded score. Here, we found the same overall trends as shown in Figure 7, with only a marginal difference between the intermediate dose scores, while the scores for the ‘rescued’ and ‘not rescued’ were the same in both cases. We have not amended the data in Figure 7B but have mentioned our method of scoring in the Materials and methods (subsection “Whole-mount in situ hybridisation analysis of gene expression”).

12) It is not clear what the two sets of graphs are in Figure 7A – are these replicates?

The two sets of graphs record different measurements, but were scored on the same set of embryos. We have now made these two panels separate to make this clearer. The first panel (Figure 7A) is scored for the number of projections that have fused to form a pillar (i.e. morphological rescue of the inner ear defect, not just reduced *vcanb* expression). Figure 7B shows the overall score of the *vcanb* staining as defined in Figure 3A. We have modified the figure legend to make this clearer and explanation in the text is found in the first paragraph of the subsection “Nifedipine, cilnidipine, tracazolate hydrochloride and FPL 64176 rescue otic defects in *adgrg6^tb233c^*mutants in a dose-dependent manner”.